# The RNA helicase UPF1 associates with mRNAs co-transcriptionally and is required for the release of mRNAs from gene loci

Anand K Singh[1], Subhendu Roy Choudhury[1], Sandip De[1†], Jie Zhang[2], Stephen Kissane[1], Vibha Dwivedi[1], Preethi Ramanathan[1], Marija Petric[1], Luisa Orsini[1], Daniel Hebenstreit[2], Saverio Brogna[1*]

[1]School of Biosciences, University of Birmingham, Birmingham, United Kingdom; [2]Life Sciences, University of Warwick, Coventry, United Kingdom

**Abstract** UPF1 is an RNA helicase that is required for nonsense-mediated mRNA decay (NMD) in eukaryotes, and the predominant view is that UPF1 mainly operates on the 3'UTRs of mRNAs that are directed for NMD in the cytoplasm. Here we offer evidence, obtained from *Drosophila*, that UPF1 constantly moves between the nucleus and cytoplasm by a mechanism that requires its RNA helicase activity. UPF1 is associated, genome-wide, with nascent RNAs at most of the active Pol II transcription sites and at some Pol III-transcribed genes, as demonstrated microscopically on the polytene chromosomes of salivary glands and by ChIP-seq analysis in S2 cells. Intron recognition seems to interfere with association and translocation of UPF1 on nascent pre-mRNAs, and cells depleted of UPF1 show defects in the release of mRNAs from transcription sites and their export from the nucleus.

DOI: https://doi.org/10.7554/eLife.41444.001

*For correspondence:
s.brogna@bham.ac.uk

Present address: †Division of Developmental Biology, Eunice Kennedy Shriver National Institute of Child Health and Human Development, National Institutes of Health, Bethesda, United States

Competing interests: The authors declare that no competing interests exist.

## Introduction

UPF1 (UP-Frameshift-1) is a universally conserved eukaryotic protein that was first identified in a *Saccharomyces cerevisiae* genetic screen for mutations that enhance up-frameshift tRNA suppression (*Culbertson et al., 1980*; *Leeds et al., 1992*), and gained other names – including *NAM7* (*S. cerevisiae*) and *SMG2* (*Caenorhabditis elegans*) – from other genetic screens (*Altamura et al., 1992*; *Hodgkin et al., 1989*; *Pulak and Anderson, 1993*). Cells that lack active UPF1 accumulate mRNAs with nonsense, frameshift or other mutant alleles that introduce a premature translation termination codon (PTC) (*Leeds et al., 1991*; *Pulak and Anderson, 1993*).

These observations are generally interpreted as evidence that UPF1 and related proteins are primarily required for nonsense-mediated mRNA decay (NMD), a conserved eukaryotic mRNA surveillance mechanism that detects and destroys mRNAs on which translation terminates prematurely (*Fatscher et al., 2015*; *He and Jacobson, 2015*; *Karousis et al., 2016*; *Kurosaki and Maquat, 2016*). NMD is mainly regarded as a quality control mechanism that prevents cells from wastefully making truncated (and potentially toxic) proteins and that regulates the selective expression of specific mRNA isoforms during cell homeostasis and differentiation (*Goetz and Wilkinson, 2017*; *Lykke-Andersen and Jensen, 2015*).

The exact role of UPF1 in NMD is uncertain though. Standard models postulate that UPF1 monitors translation termination on ribosomes by interacting with a peptide release factor (eRF1 or eRF3). However, recent reports on mammalian translation systems have suggested, in contrast to earlier reports on other organisms (*Czaplinski et al., 1998*; *Ivanov et al., 2008*; *Kashima et al.,*

*2006*; *Keeling et al., 2004*; *Singh et al., 2008*; *Wang et al., 2001*), that UPF1 does not bind to either of these. They suggested, instead, that UPF3B may contact release factors, slow the termination of translation and facilitate post-termination release of ribosomes – and so fulfil the termination monitoring role that has been assigned to UPF1 (*Neu-Yilik et al., 2017*).

UPF1 is an ATP-driven helicase that unwinds RNA secondary structures and so can displace RNA-bound proteins (*Bhattacharya et al., 2000*; *Chakrabarti et al., 2011*; *Czaplinski et al., 1995*; *Fiorini et al., 2015*; *Franks et al., 2010*). UPF1 is predominantly associated with 3'UTRs of cytoplasmic mRNAs which indicates that it might be selectively recruited to or activated on NMD targets with abnormally long 3'UTRs (*Karousis et al., 2016*; *Kurosaki and Maquat, 2016*). However, UPF1 appears to bind mRNAs fairly indiscriminately, regardless of the position of the stop codon or the PTC and whether or not the mRNA possess NMD-inducing features such as an abnormally long 3'UTR or an exon junction downstream of the stop codon (*Hogg and Goff, 2010*; *Hurt et al., 2013*; *Zünd et al., 2013*). UPF1 helicase activity is required for NMD, but how it helps to target particular transcripts for NMD remains unclear. In view of these and other perplexing observations, it has also been questioned whether cells do possess any such mechanism to discriminate between PTCs and normal stop codons (*Brogna et al., 2016*).

UPF1 is most abundant in the cytoplasm where its roles discussed above depend on ribosomal translation and occur on cytoplasmic mRNAs. A fraction of UPF1 was expected in the nucleus though, as the protein traffics in and out of the nucleus in mammalian cells (*Ajamian et al., 2015*; *Mendell et al., 2002*). Some studies have concluded that within the nucleus UPF1 plays a distinct and direct role in DNA replication, which would be unrelated to gene expression (*Azzalin and Lingner, 2006*; *Azzalin et al., 2007*; *Carastro et al., 2002*; *Chawla et al., 2011*). However, the negative effects that depletion of UPF1 has on DNA replication and cell division could be an indirect consequence of NMD suppression altering the expression of genes required in such processes (*Varsally and Brogna, 2012*). Moreover, there is evidence that nuclear UPF1 might contribute directly to RNA processing, at least in specific instances, and is required for nuclear export of HIV-1 genomic RNA in HeLa cells (*Ajamian et al., 2015*; *de Turris et al., 2011*; *Flury et al., 2014*; *Varsally and Brogna, 2012*). Additionally, CLIP data indicate direct binding of UPF1 with the abundant nuclear-localised metastasis-associated lung adenocarcinoma transcript 1 (MALAT1) in mammalian cells (*Zünd et al., 2013*).

In the present study, we show direct evidence that UPF1 is globally involved in nuclear processing of mRNAs in *Drosophila*. First, we demonstrate that UPF1 is a highly mobile protein that constantly shuttles between the nucleus and cytoplasm, and its distribution in the cell, with more in the cytoplasm than the nucleus, depends on its RNA binding properties and approximately reflects that of mRNA. UPF1 associates with nascent transcripts on chromosomes – mostly with Pol II transcribed, but also with some Pol III-transcribed genes. Relatively more of the transcript-associated UPF1 is bound with exons than with introns, suggesting that intron recognition might act as a roadblock to the 5'-to-3' transit of UPF1 along the pre-mRNA. Most strikingly, UPF1 is needed for the efficient release of polyadenylated mRNAs from most chromosomal transcription sites and for their export from the nuclei. These observations indicate that UPF1 starts scanning pre-mRNA transcripts whilst they are still being assembled into ribonucleoprotein (RNP) complexes on chromosomes and suggest that UPF1 fulfils previously unrecognised role(s) in facilitating nuclear processes of gene expression and mRNA export. The broad and dynamic association of UPF1 with mRNAs redefines it from being primarily an NMD-inducing factor to being a global player in mRNA processing in the nucleus as well as in the cytoplasm, and might also explain why none of the prevailing models satisfactorily explain how UPF1 could target specific transcripts to NMD.

## Results

### Drosophila anti-UPF1 antibodies

To explore the functions of UPF1, we generated three monoclonal anti-peptide antibodies that target regions of *Drosophila* UPF1 outside the RNA helicase domain: one epitope in the N-terminal flanking regions (antibody 1C13 against Pep2), and two near the C-terminus (Ab 7D17 *vs.* Pep11; and Ab 7B12 *vs.* Pep12) (see *Figure 1—figure supplement 1A and B* and *Supplementary file 1*). Following purification from hybridoma supernatants, each antibody detected UPF1 as a single band

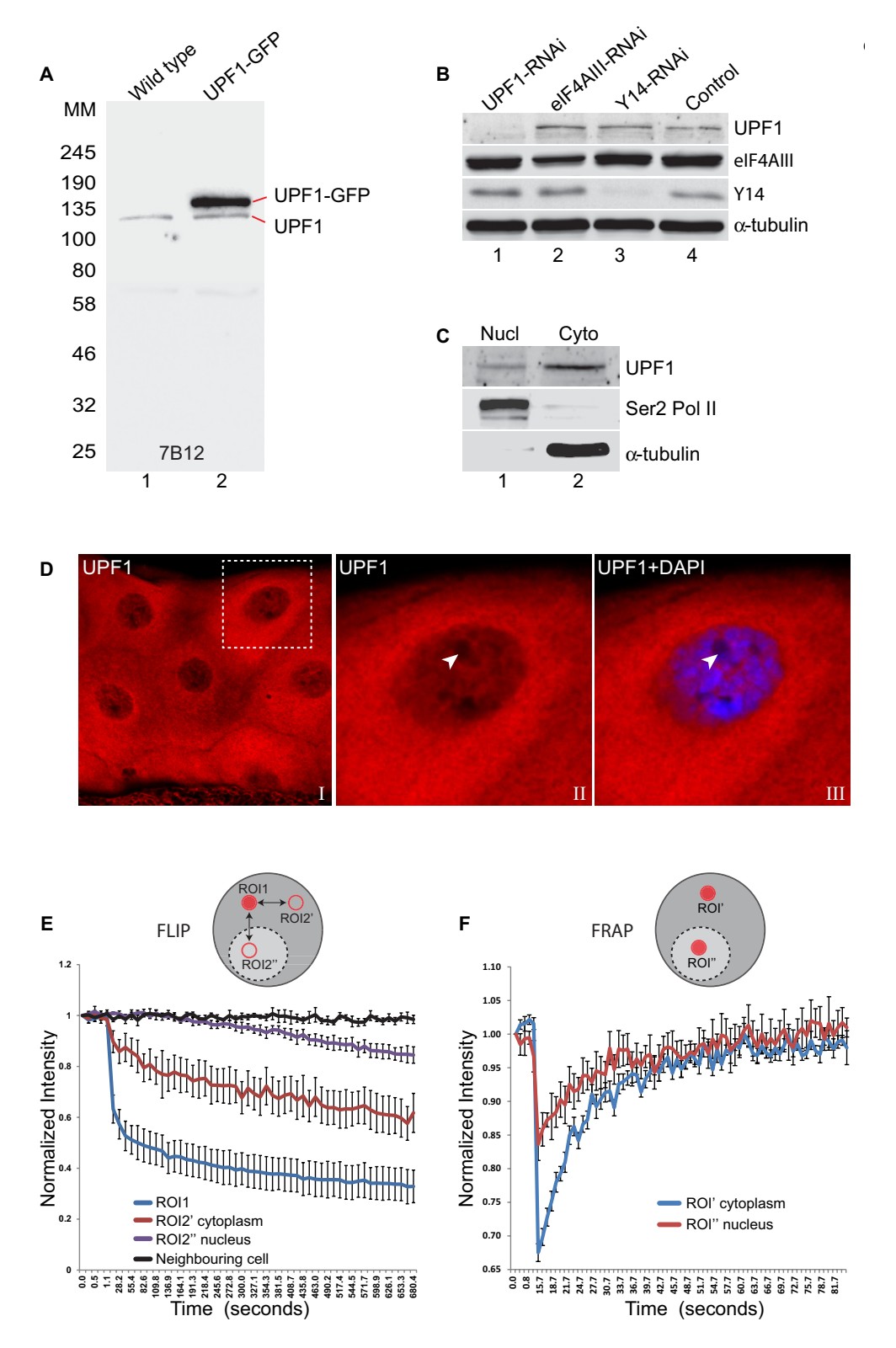

**Figure 1.** UPF1 continuously shuttles between nucleus and cytoplasm. (**A**) Western blotting of whole-cell lysate from either normal (lane 1) or transfected S2 cells expressing UPF1-GFP (lane 2), probed with the UPF1 monoclonal antibody 7B12. The proteins run according to their expected molecular weights: UPF1 (~130 kDa), and UPF1-GFP (~157 kDa) (**B**) Western blotting of S2 cells treated with dsRNA targeting UPF1 or other RNA binding proteins indicated, used as controls. Different sections of the membrane were probed with anti-UPF1 (7B12, top row), anti-eIF4AIII (row 2), anti-

*Figure 1 continued on next page*

*Figure 1 continued*

Y14 (row 3) or anti-α-tubulin (row 4) as a loading control. (C) Western blotting of UPF1 following nuclear (Nucl) and cytoplasmic (Cyto) fractionation of S2 cells. RNA Pol II and α-tubulin were detected using the corresponding antibodies on the same blot (shown below). (D) Fluorescence immunolocalisation of UPF1 (Cy3, red) in third instar larval salivary gland. The arrowheads in panel II and III (magnified view of boxed area in panel I) point to the nucleolus, identified by no DAPI staining, which, as other nucleoli, shows no UPF1 signal in its centre. (E) The plot shows fluorescence loss in photobleaching (FLIP) of GFP-UPF1in salivary gland cells photobleached in ROI1 (red circle, cytoplasm) and then GFP signal measured at the identical time points in two separate ROI2s (red rings), in either cytoplasm or nucleus; both equidistant from ROI1. The different lines show rate of GFP fluorescence loss in either the photobleached ROI1 (blue line), or ROI2' in the cytoplasm (red line) or ROI2'' in nucleus (purple line). Change in fluorescence intensity at equivalent regions in neighbouring cells was measured as a control during the same time-course (black line). Y-axis shows normalised relative fluorescence intensity while X-axis shows time (seconds) from the start of imaging. Quantification based on imaging experiments in eight different cells. (F) Plot shows fluorescence recovery after photobleaching (FRAP) of GFP-UPF1 in either cytoplasm (ROI', blue line) or nucleus (ROI'', red line) of salivary gland cells. Line values represent the average of eight separate measurements in different cells. Error bars in E and F indicate ±Standard Error.

DOI: https://doi.org/10.7554/eLife.41444.002

The following figure supplements are available for figure 1:

**Figure supplement 1.** Generation of monoclonal antibodies against Drosophila UPF1.

DOI: https://doi.org/10.7554/eLife.41444.003

**Figure supplement 2.** UPF1 immunostaining signals are reduced in UPF1-RNAi salivary glands.

DOI: https://doi.org/10.7554/eLife.41444.004

**Figure supplement 3.** UPF1 subcellular localisation in different larval tissues.

DOI: https://doi.org/10.7554/eLife.41444.005

by western blotting of *Drosophila* S2 cell extracts, with minimal cross-reactivity to other proteins (*Figure 1A*, *Figure 1—figure supplement 1C and D*). The antibodies also detected a second, larger band of the expected molecular weight in extracts from transfected S2 cells that over-express UPF1-GFP. Unless otherwise indicated, antibody 7B12 was used in the experiments described below. As expected, UPF1-RNAi specifically reduced the amount of UPF1 in S2 cells without affecting the levels of several other proteins we tested as controls (*Figure 1B*).

## UPF1 rapidly shuttles between nucleus and cytoplasm

We examined the subcellular localisation of immunostained UPF1 in *Drosophila* salivary glands, which are made up of large secretory cells with polytene nuclei. UPF1 was most abundant in the cytoplasm and perinuclear region, and there was also distinct but less intense nuclear staining, mainly around the chromosomes and around the nucleolus (*Figure 1D*). A similar subcellular distribution of UPF1 was detected with the remaining two antibodies tested in wild-type salivary glands; and, the signal was drastically reduced in UPF1-RNAi glands, consistent with all three antibodies being specific (*Figure 1—figure supplement 2A*). Following cell fractionation of S2 cells, α-tubulin and RNA Pol II were, as expected, restricted to the cytoplasmic and nuclear fractions, respectively – and a small proportion of the UPF1 co-purified with nuclei whilst most was in the cytoplasmic fraction (*Figure 1C*), consistent with the subcellular localisation detected by immunostaining.

UPF1 was also detected both in the cytoplasm and in the nucleus in other larval tissues, with varying immunostaining intensities. Perinuclear UPF1 was more apparent in salivary glands that are at a later stage of development (*Figure 1—figure supplement 2A*); and it was also obvious in Malpighian tubules and gut, which also showed an increased intra-nuclear UPF1 signal (*Figure 1—figure supplement 3*). In enterocytes (EC), staining was similar between the cytoplasm and the nucleus, while the most intense UPF1 signal was perinuclear (*Figure 1—figure supplement 3B*). The perinuclear presence was also apparent in salivary glands expressing GFP-UPF1, where UPF1 co-localised with binding of wheat germ agglutinin (WGA) – a lectin that predominantly interacts with O-GlcNAc-modified nuclear pore proteins (*Mizuguchi-Hata et al., 2013*) (*Figure 1—figure supplement 2B*).

Since UPF1 is present both in the cytoplasm and nuclei, with relative quantities varying between cell-types, we wondered how rapidly UPF1 shuttles between cell compartments. We used two live cell imaging techniques – Fluorescence Loss in Photo-bleaching (FLIP) and Fluorescence Recovery after Photo-bleaching (FRAP) (*Singh and Lakhotia, 2015*) – to examine the mobility of GFP-UPF1 in salivary gland cells. FLIP revealed that sustained photobleaching of a small area of the cytoplasm led, within the continuously illuminated area, to an initial rapid decrease in GFP-UPF1fluorescence

followed by a continued slower reduction. Fluorescence also declined steadily both elsewhere in the cytoplasm, and, more slowly, within the nucleus (*Figure 1E*). These observations demonstrate ongoing diffusion of UPF1 throughout the cytoplasm, and that nuclear UPF1 can leave the nucleus and enter the photodepletable cytoplasmic UPF1 pool at a fairly steady rate. The FRAP studies monitored the speed with which unbleached GFP-UPF1 diffuses into and repopulates a photobleached region of the cytoplasm or nucleus. Almost all of the UPF1 in each cell compartment was rapidly mobile, with the halftime for repopulation of each bleached area being only a few seconds (*Figure 1F*).

## UPF1 shuttling between the nucleus and the cytoplasm requires its RNA helicase activity

Nucleo-cytoplasmic shuttling has been reported in HeLa cells, with UPF1 accumulating in the nuclei following treatment with leptomycin B (LMB) (*Mendell et al., 2002*), a drug that selectively inhibits CRM1-mediated protein export from the nucleus in most eukaryotes (*Fukuda et al., 1997*). We therefore explored the intracellular localisation and dynamics of a GFP-tagged UPF1 in *Drosophila* salivary glands; this (GFP-UPF1) showed a intracellular distribution similar to that of the endogenous protein, with an intense cytoplasmic signal and a weaker, but still obvious, signal in the regions occupied by the chromosomes (*Figure 2A*: left panel, the cytoplasmic texture of the salivary gland cells in these confocal images reflects the fact that they are packed with secretory vesicles at this stage of larval development). In glands treated with LMB for 60 min most of the GFP-UPF1 was observed within the nucleus, being largely excluded from the nucleolus (*Figure 2A*, right panels), suggesting that UPF1 exits from the nucleus, directly or indirectly, via a CRM1-dependent mechanism. This UPF1 redistribution was rapid in living glands: UPF1 was accumulating in the nucleus by the earliest time we could collect images (within ~5–6 min from adding LMB), and much of the cell's UPF1 localised in the nucleus within half an hour (*Figure 2—figure supplement 1A*).

Heat-shock caused a similar redistribution of much of UPF1 from the cytoplasm to the nucleus (*Figure 2—figure supplement 1B*, left panel), which was partially reversed when the tissue was returned to its normal temperature (*Figure 2—figure supplement 1B*, right panel).

Next, we examined whether the shuttling of UPF1 requires its RNA helicase activity in S2 cells transfected with constructs expressing either the wild-type or a mutant version of UPF1 with two amino acid substitutions (DE617AA) that inhibit its RNA helicase activity due to the loss of ATP hydrolysis (*Bhattacharya et al., 2000*). Both the wild type and the mutant proteins, tagged with GFP at either the N- or the C-terminal, were more abundant in the cytoplasm, as expected (*Figure 2B*, left panel). A portion of UPF1, the DA617AA mutant in particular, localised in bright fluorescent dots in the cytoplasm, possibly corresponding to P bodies (*Brogna et al., 2008*). However, whilst the wild-type UPF1 relocalised to the nucleus following the LMB treatment, resulting in more UPF1 present in the nucleus than in the cytoplasm, the distribution of the DE617AA mutants was unaffected (*Figure 2B*, right panel).

The data indicate that wild-type UPF1 is freely mobile within cell compartments and that it constantly moves in and out of the nucleus by mechanisms that involve the CRM1-dependent nuclear export pathway and requires its RNA helicase activity.

## UPF1 associates with transcribing regions of the chromosomes

To gain insight into the role(s) of UPF1 in the nucleus, we used immunostaining to examine whether it associates with the polytene chromosomes of *Drosophila* salivary glands. These well-characterised giant interphase chromosomes are formed after multiple rounds of endoreplication without chromosomal segregation, and they provide a powerful system to visualise transcription and pre-mRNA processing at individual gene loci.

UPF1 was present predominantly at interbands and puffs: cytologically distinct chromosome regions in which the chromatin is less condensed and that correspond to transcriptionally active sites (*Figure 3A*). The immunofluorescence signal appears to be specific, as: (a) UPF1-RNAi drastically depletes the endogenous UPF1 chromosomal signal (*Figure 3—figure supplement 1A* and *Figure 3—figure supplement 1B*); (b) the other two UPF1 antibodies produced a similar immunostaining banding pattern (*Figure 3—figure supplement 1C*); and, (c) transgenically over-expressed

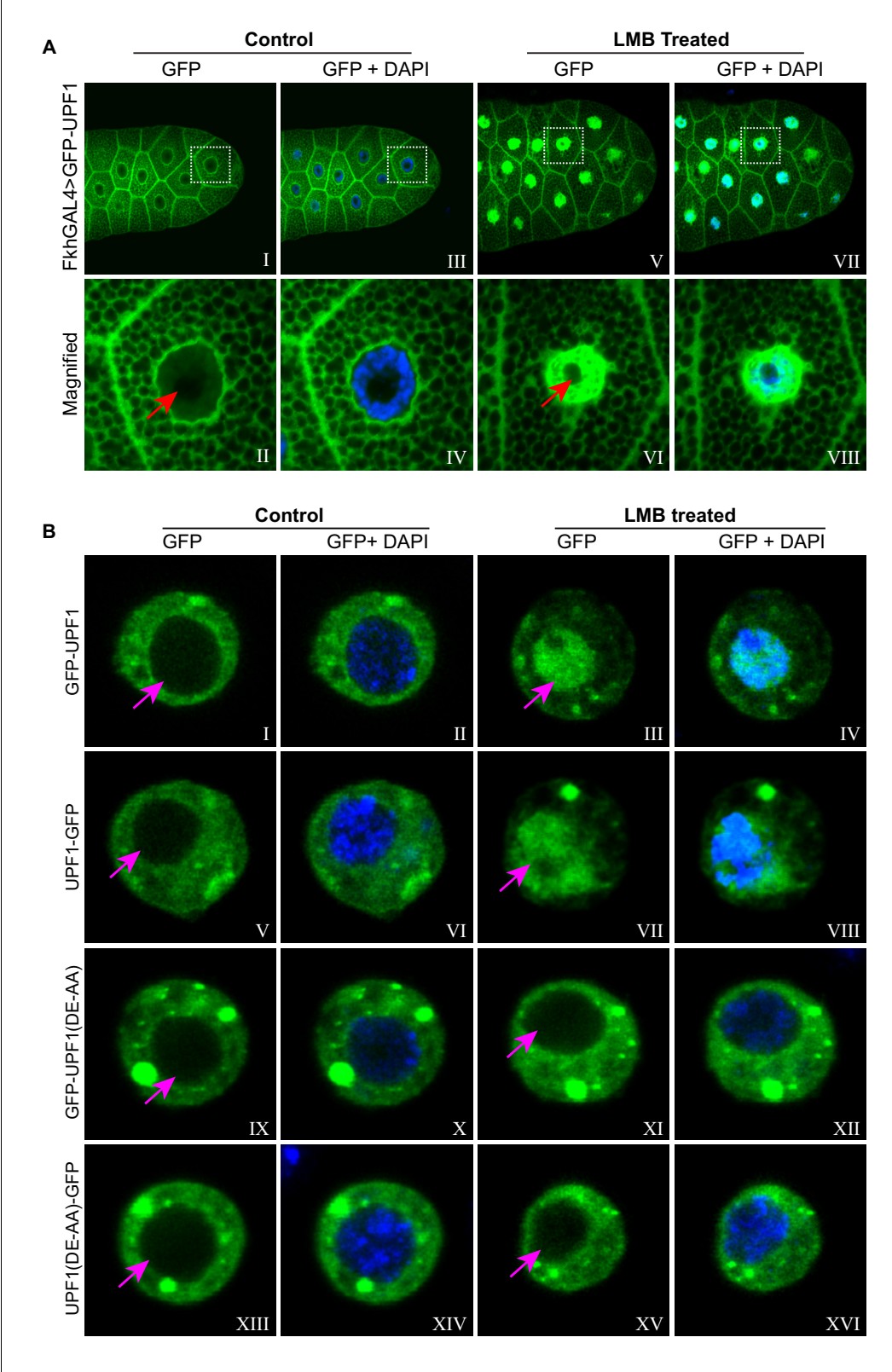

**Figure 2.** UPF1 shuttling between nucleus and cytoplasm requires its RNA helicase activity. (**A**) Imaging of third instar larval salivary glands over-expressing GFP-UPF1 (green), incubated for 1 hr in either normal M3 media (Control, **I to IV**) or supplemented with 50 nM LMB (LMB, **V to VIII**). Panels II, IV, VI and VIII are magnified views of the boxed areas in panel I, III, V and VII, respectively. The red arrows in II and VI indicate the nucleoli. Nuclei were counter-stained with DAPI (blue). (**B**) Imaging of transfected Drosophila S2 cells expressing either GFP-UPF1 (**I to IV**), UPF1-GFP (**V to VIII**), GFP-

*Figure 2 continued on next page*

Figure 2 continued

UPF1(DE-AA) (IX to XII) or UPF1(DE-AA)-GFP (XIII to XVI), incubated for 1 hr with or without 50 nM LMB (right vs. left panels). The magenta coloured arrows indicate the nuclei, which were counter-stained with DAPI (blue) in the even numbered panels.

DOI: https://doi.org/10.7554/eLife.41444.006

The following figure supplement is available for figure 2:

**Figure supplement 1.** UPF1 is highly dynamic within both nucleus and cytoplasm.

DOI: https://doi.org/10.7554/eLife.41444.007

UPF1-GFP, detected either by its fluorescence or with an anti-GFP antibody, also shows a similar banding pattern on the chromosomes (*Figure 3—figure supplement 1D*).

We then undertook double immunostaining of chromosomes for UPF1 and for Ser2 Pol II – the form of Pol II that transcribes through the main body of genes which is characterised by having the C-terminal domain (CTD) of its largest subunit Ser2-phosphorylated (*Boehm et al., 2003*). Much of the UPF1 co-localised with Ser2 Pol II, as would be expected from this type of banding pattern (*Figure 3—figure supplement 2A*).

The association of UPF1 with the chromosomes depends on transcription. This is illustrated by the changes in UPF1 immunostaining that followed heat-shock, which induces transcription at specific cytological puffs encoding heat-shock proteins and of hsrω lncRNAs at locus 93D (*Lakhotia et al., 2012*). This revealed a pattern of UPF1 association at heat shock puffs and of detachment from most other transcription sites (*Figure 3B*). UPF1 was recruited to activated heat-shock genes that either contained (33B, 63B, 64F, 67B, 70A and 93D) or lacked (87A, 87C and 95D) introns (*Figure 3B*).

These observations suggested that UPF1 associates with genes that are being transcribed. UPF1 was also recruited to other genes following transcription activation, such as to an ecdysone-inducible transgene (S136 at chromosomal position 63B), at normal temperature (*Choudhury et al., 2016*). No UPF1 was found at this locus on the wild-type chromosome, but UPF1 was clearly associated with the transcription puff which was produced at this location following ecdysone activation of the transgene (*Figure 3C*).

## UPF1 mainly associates with Pol II transcription sites and depends on the nascent transcript

We examined UPF1 and Ser2 Pol II association with multiple gene loci by chromatin immunoprecipitation (ChIP) of S2 cell extracts, followed by high-throughput DNA sequencing (ChIP-Seq). UPF1 was associated with many transcriptionally active genes, most of which are Pol II transcription sites. *Figure 4A* shows enrichment profiles of UPF1 and of Ser2 Pol II along a representative chromosome region. *Actin5C* provided a striking example of correspondence between the ChIP-seq and polytene immunostaining results: it was one of the most UPF1-enriched genes in the ChIP-seq data (*Supplementary file 2*, *Figure 5—figure supplement 1B* shows the UPF1 ChIP-seq profile of *Actin5C*) and displayed one of the brightest UPF1 chromosomal signals at the gene locus corresponding to interband 5C on the X chromosome (*Figure 3A*). The ChIP-seq data also show UPF1 association with a few Pol III genes (*Supplementary file 2*, to be discussed later).

The enrichment profile of UPF1 at Pol II loci closely followed that of Ser2 Pol II, and UPF1 enrichment being the greatest at highly expressed genes (*Figure 4A*; *Figure 4—figure supplement 1A* and *Figure 5—figure supplement 1* show additional examples of UPF1-enriched genes). A close correlation was observed between UPF1 and Ser2 Pol II ChIP-seq signals, and also between UPF1-ChIP signals and mRNA levels (*Figure 4B and C*). Real-time PCR was used to validate the ChIP-seq data at several genes, both in S2 cells and salivary glands (*Figure 4—figure supplement 2B and C*; other examples are shown below). UPF1-RNAi led to a reduction in UPF1 enrichment at transcription sites, both confirming the specificity of the antibody and validating the ChIP protocol (*Figure 4—figure supplement 2C*).

A metagene analysis of the ChIP-seq data shows that UPF1 is associated with genes, particularly with highly expressed genes (blue trace), and throughout their transcription units (*Figure 4D*), whereas Ser2 Pol II typically shows higher loading around transcription start sites (TSS) – corresponding to promoter-proximal Pol II pausing sites, as previously reported in *Drosophila* and other

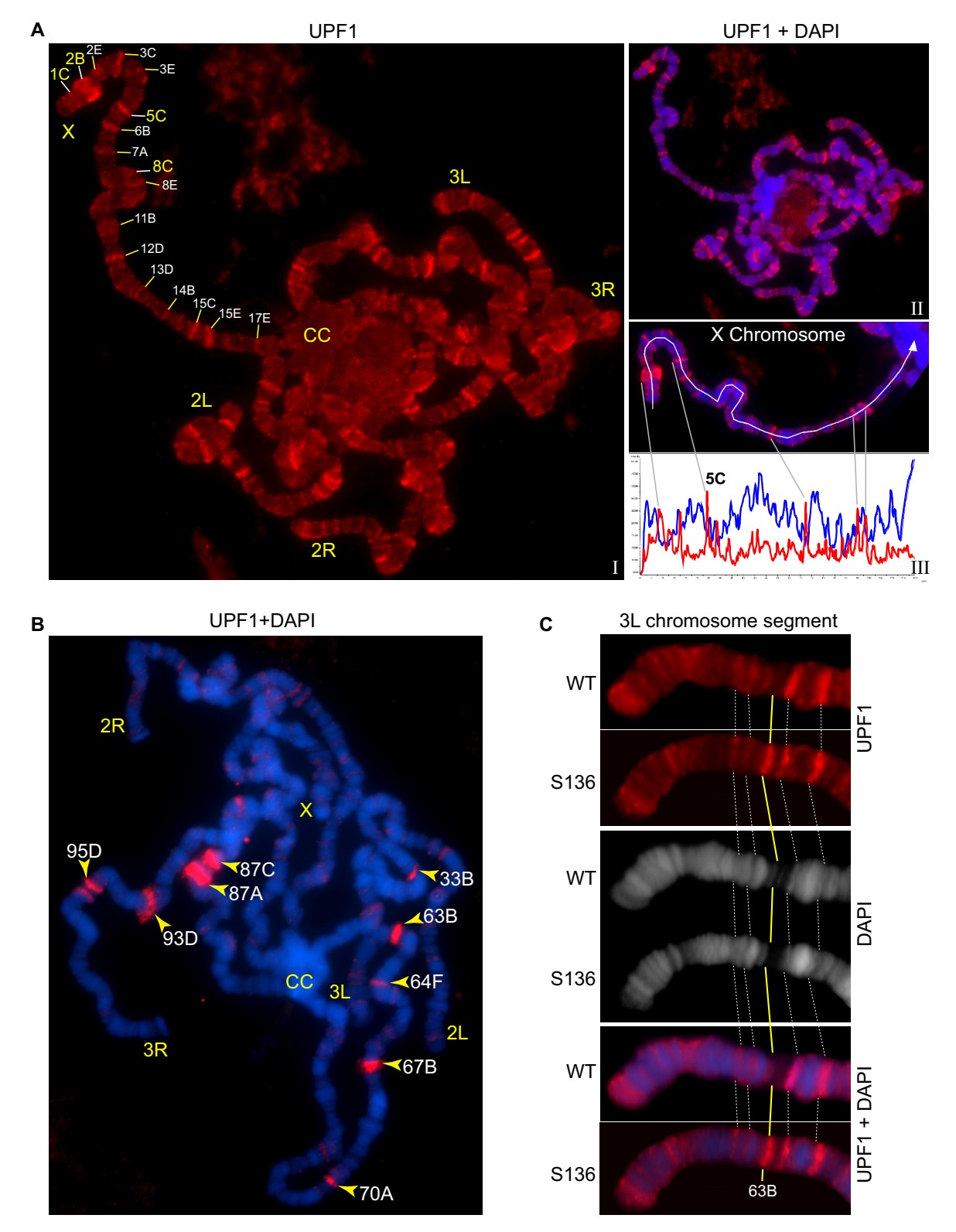

**Figure 3.** UPF1 binds at transcriptionally active sites on the polytene chromosomes. (**A**) Fluorescence immunolocalisation of UPF1 (Cy3, red, (I) on polytene chromosomes (DAPI, blue, II). Chromosome arms (X, 2L, 2R, 3L and 3R) and chromocentre (CC) are labelled. The labels indicate cytological locations of interband regions at the X chromosome, presenting apparent UPF1 signal. The line profile (III, white panel) shows signal intensities along the white line drawn on the X chromosome, UPF1 (red) and DAPI (blue). (**B**) Immunolocalisation of UPF1 (red) on polytene chromosomes derived from

*Figure 3 continued on next page*

Figure 3 continued

larvae subjected to a 40 min heat shock at 37°C. UPF1 signals are primarily detected at heat shock gene loci, indicated by their cytological locations (yellow arrowheads), using their standard nomenclature. (C) Immunolocalisation of UPF1 (red) at an ecdysone induced transgene (named S136) located at cytological position 63B (yellow line) and the same region on the wild type chromosome after ecdysone treatment. The white dotted lines indicate flanking bands as mapping reference. Chromosomes were stained with DAPI (grey in middle panel or blue in bottom panel).

DOI: https://doi.org/10.7554/eLife.41444.008

The following figure supplements are available for figure 3:

**Figure supplement 1.** RNAi fully depletes UPF1 signal at the salivary gland polytene chromosomes.

DOI: https://doi.org/10.7554/eLife.41444.009

**Figure supplement 2.** UPF1 chromosomal association is transcription and nascent RNA dependent.

DOI: https://doi.org/10.7554/eLife.41444.010

organisms (*Adelman and Lis, 2012*; *Muse et al., 2007*). Typically, most gene-associated UPF1 was further downstream than the TSS-proximal Ser2 Pol II peak, especially at highly expressed genes (*Figure 4E*). Striking examples of this pattern are the *NAT1* and *Su(z)2* genes (*Figure 4A*) and the *α-Tub84B* gene (*Figure 4—figure supplement 1A*, including some of the gene described further below.

A comparison of the UPF1 loading of genes with different Ser2 Pol II loading profiles suggests that UPF1 association depends on transcription elongation: UPF1 did not associate with genes at which Ser2 Pol II was associated only with the TSS pausing site and which were not being actively transcribed (*e.g. Adam TS-A*, panel five in *Figure 4—figure supplement 1A*).

The association of UPF1 with Pol II transcription sites is partially sensitive to RNase treatment, suggesting that UPF1 binds nascent RNA. This was apparent both for immunostained UPF1 on polytene chromosomes *Figure 3—figure supplement 2B and C*) and when assayed by ChIP/qPCR at specific genes in S2 cells (*Figure 4—figure supplement 2D*). UPF1 association was, though, less sensitive to RNase treatment than that of the RNA binding protein hnRNPA1 (*Figure 3—figure supplement 2B and C*), which is almost completely detached from the chromosome following the same RNase treatment. Some of UPF1 co-purifies with Ser2 Pol II in a standard immunoprecipitation of S2 nuclear cell extracts, the interaction being again sensitive to RNase treatment (*Figure 4—figure supplement 2E*): less than that of hnRNPA1, but comparable to that of eIF4AIII, one of the exon junction complex (EJC) proteins that are loaded onto nascent RNAs (*Choudhury et al., 2016*).

We also examined the effect of 5,6-dichloro-1-beta-D-ribofuranosylbenzimidazole (DRB) on salivary glands, a drug that blocks Pol II transcription by inhibiting Ser2 phosphorylation (*Bensaude, 2011*). In the presence of DRB, unphosphorylated Pol II (Pol II) initiates transcription but does not engage in productive elongation as this would require Ser2-phosphorylated Pol II (Ser2 Pol II) (*Adelman and Lis, 2012*). DRB treatment, as expected, left interbands and puffs cytologically unaffected, however, it markedly reduced the amount of UPF1 associated with gene loci (*Figure 3—figure supplement 2D and E*), providing further evidence that transcript elongation into the body of the gene is needed for this association to occur. DRB also reduced the association of UPF1 and Ser2 Pol II with genes, such as the highly expressed *RpL23A*, in S2 cells (*Figure 4F*).

## UPF1 at Pol III transcription sites

UPF1 was found mainly at Pol II transcription sites, most of which are protein-coding genes, however, our ChIP-seq data revealed that UPF1 also binds at some Pol III genes. The latter included 7SK and both paralogous genes of 7SL snRNAs (*Figure 4—figure supplement 1B*) – but not, for example, the much more numerous Pol III-transcribed tRNA genes (*Figure 4—figure supplement 1C*, *Supplementary file 2*).

## Intron recognition interferes with UPF1 nascent transcript association

UPF1 was recruited both to intron-containing and intronless genes that were undergoing transcription (*Figure 5—figure supplement 1*, and see also the earlier discussion of heat-shock gene activation), so recruitment did not depend on pre-mRNA splicing. Within intron-containing genes, however, more UPF1 was associated with exons than with introns – as can be seen in the ChIP-seq profiles of highly UPF1-enriched genes such as *Xrp1* (*Figure 5A*; and *Figure 5—figure supplement 1* shows other examples of genes displaying this pattern). Additionally, it appears that relatively

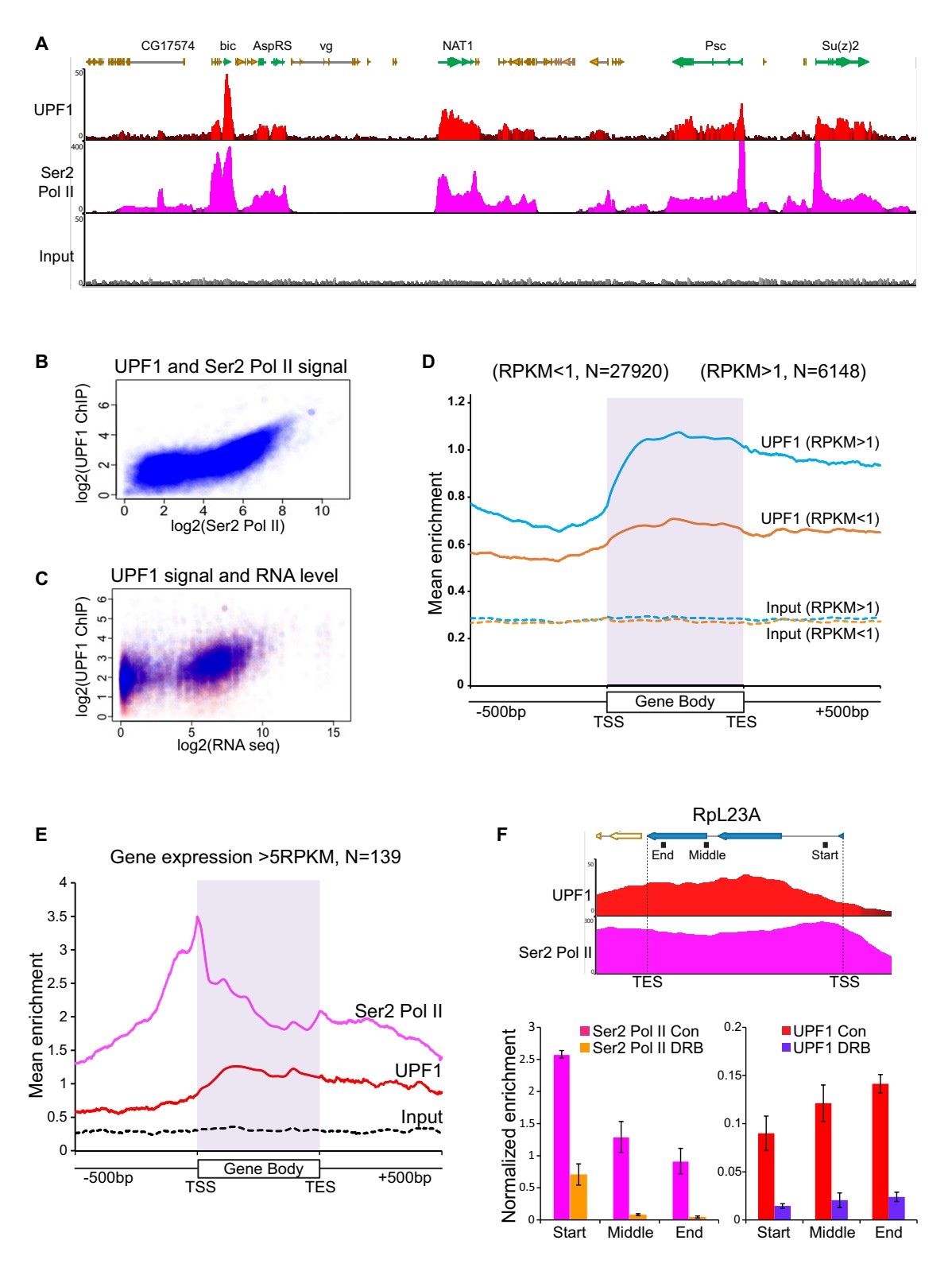

**Figure 4.** UPF1 associates at Pol II transcription sites. (**A**) Genome browser visualisation of UPF1 (red) and Ser2 Pol II (pink) ChIP-seq enrichment profiles at a representative chromosomal region in S2 cells, including highly active genes (green) and low or inactive genes (orange). The input profile (grey) is shown in the bottom panel on the same scale as that of UPF1 (**B**) Scatter plot showing correlation between normalised exon reads in UPF1 and Ser2 Pol II ChIP-seq samples. (**C**) Scatter plot showing relationship between normalised UPF1 ChIP-reads vs. mRNA-seq expression levels; data points

*Figure 4 continued on next page*

*Figure 4 continued*

corresponding to either exons (blue) or introns (red). Correlation values are 0.485 and 0.398, for exons (blue line) and introns (red line) respectively. (**D**) Metagene profiles showing average UPF1 occupancy at either active (blue, RPKM >1) or inactive/low expressed transcription units (RPKM <1, orange), gene body (scaled to 16 bins of gene full length) plus 500 bp from either end. The number of individual transcription units (**N**) used for this analysis is given on top. Corresponding normalised input profiles are shown by dotted lines. (**E**) Superimposed metagene plots of UPF1 (red) and Ser2 Pol II (pink) at highly expressed gene loci (RPKM >5). The input enrichment profile for same gene set is shown by the dotted line (black). (**F**) Graph shows ChIP-seq enrichment profiles of UPF1 (red) and Ser2 Pol II (pink) at the RpL23A gene. Bottom panel shows real-time PCR quantification of Ser2 Pol II (left) and UPF1 (right) average enrichment at RpL23A gene, based on two separate ChIP replicates from either normal or DRB treated S2 cells. The relative position of the three amplicons tested (Start, Middle and End of the gene) are indicated by black boxes underneath the gene schematic on top. Error bars indicate ±Standard Error.

DOI: https://doi.org/10.7554/eLife.41444.011

The following figure supplements are available for figure 4:

**Figure supplement 1.** ChIP-seq profiles of UPF1 at representative Pol II genes and some Pol III loci.

DOI: https://doi.org/10.7554/eLife.41444.013

**Figure supplement 2.** Real-time PCR validation of UPF1 ChIP association at selected genes.

DOI: https://doi.org/10.7554/eLife.41444.012

more of UPF1 is associated with downstream exons than with the first exon, at *Xrp1* as well as several of the other genes; notably, in many such cases most of the first exon corresponds to the 5'UTR (see examples in *Figure 5—figure supplement 1*).

This exon-biased UPF1 enrichment was confirmed by real time PCR in multiple ChIP experiments (at *Xrp1* shown in *Figure 5B*; and *Socs36E*, not shown); and it is genome-wide, as demonstrated by comparing UPF1 association with introns and with their flanking exons in the ChIP-seq data from many genes (*Figure 5C*), UPF1 enrichment is significantly higher for both the left (p=6.737e-8) and the right flanking exon (2.391e-9); for details of how we corrected for possible bias in chromatin fragmentation or sequencing coverage, see Materials and methods. This pattern is made visually apparent by plotting normalised enrichment in exons and introns, each scaled as a percentage of their full length (*Figure 5D*), and by comparing the density plots of normalised UPF1 enrichment values in introns and flanking exons, which show more values that are enriched in exons than introns (*Figure 5—figure supplement 2A*, compare red and yellow lines *vs.* the blue line in the right half of the graph).

The lower frequency with which UPF1 associated with introns suggested that either splicing enhances binding to downstream exons or that intron recognition interferes with the UPF1 interaction (*Figure 5—figure supplement 2B*, Model one and Model two respectively). We considered that 5' splice sites (5'ss) at the start of introns, where the initial U1 snRNP spliceosome complex would bind, might act as road-blocks to UPF1 translocation along nascent pre-mRNAs, hence, removal of U1 might allow UPF1 to move across the intron (Model 2). Consistent with this interpretation, the normal bias towards UPF1-exon association in *Xrp1* was abolished in U1-70K-depleted cells but persisted in cells depleted of eIF4AIII or Y14 (*Figure 5—figure supplement 2C*; a similar effect was observed at *Socs36E*, data not shown); Y14 and eIF4III are two EJC proteins that bind the nascent pre-mRNA but are not likely to play a direct splicing role in *Drosophila*; see (*Choudhury et al., 2016*). Moreover, genes with the most prominent exon-biased UPF1 enrichment, such as *Xrp1*, are efficiently co-transcriptionally spliced (see the Nascent RNA-seq profile in *Figure 5A*), whereas genes with no detectable exon-biased UPF1 enrichment, such as *CG5059*, are poorly co-transcriptionally spliced (*Figure 5—figure supplement 1C*) and are typically expressed at low levels, as reported (*Khodor et al., 2011*). It seems therefore, that intron recognition interferes with the association of UPF1 with the unspliced nascent transcript.

## UPF1 depletion leads to nuclear mRNA retention

We also assessed whether depleting UPF1 in the salivary gland cells of third instar larvae would have any effect on mRNA release from transcription sites and its subsequent processing and export from the nucleus.

First we examined the overall cellular distribution of poly(A) RNA – which is referred to simply as poly(A) – by oligo(dT) FISH (fluorescence in situ hybridisation): this should detect mRNA that has been transcribed, spliced, released from Pol II and polyadenylated. In wild-type cells poly(A) was

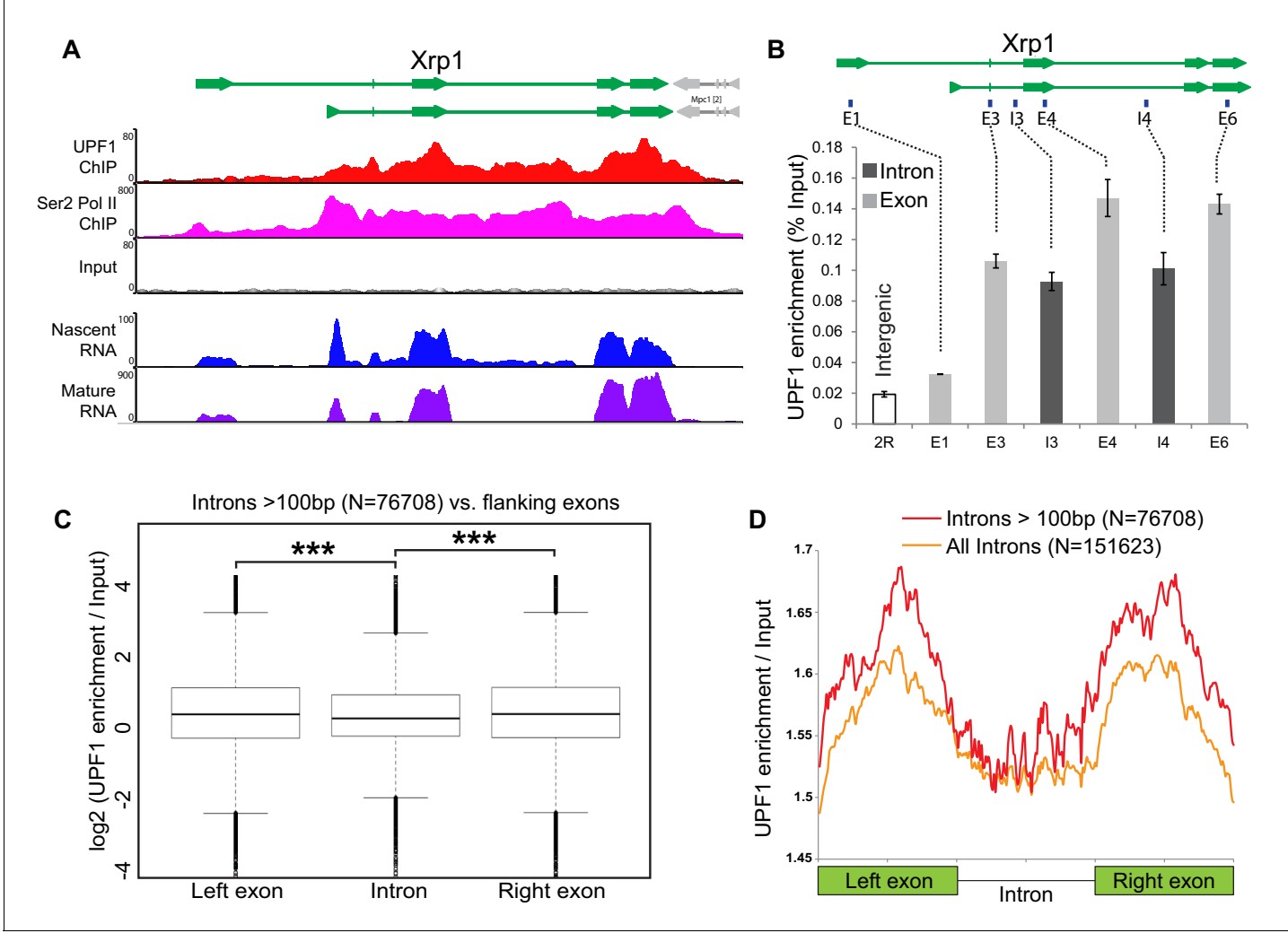

**Figure 5.** Intron recognition interferes with UPF1 association on nascent transcripts. (**A**) Schematic of the *Xrp1* locus (top) showing its two main transcription units. Below, UPF1 (red) and Ser2 Pol II (pink) ChIP-seq profiles at this gene; that of the input is shown below (grey). The bottom two panels show nascent RNA-seq (blue) and poly(A) RNA-seq (purple) profiles. (**B**) Real-time PCR quantification of average enrichment in different regions in either exons (**E1, E3, E4 and E6**) or introns (**I3 or I4**) in multiple UPF1 ChIP replicates. Error bars indicate ±Standard Error. (**C**) Box plots of normalised UPF1 ChIP-seq reads mapping at either left exon (shown on left), intron (middle) or right exon (on right). Whiskers correspond to ±1.5 interquartile range with respect to quartiles. Wilcoxon rank sum test values are: left exon vs. intron, p-value=6.737e-08; right exon vs intron, p-value=2.391e-09; and, left exon vs. right exon p-value=0.606. ***p<0.001 for difference in UPF1 signal between intron and its flanking exon. (**D**) Line profile of average UPF1 ChIP-seq/input enrichment expressed as percentage of full length in either exons or intron. Analysis is based on 151623 introns of any length (orange line) or 76708 introns longer than 100 bp (red line) as annotated in the dm6 genome release.

DOI: https://doi.org/10.7554/eLife.41444.014

The following figure supplements are available for figure 5:

**Figure supplement 1.** Additional examples of UPF1 ChIP-seq profiles at genes with or without introns.
DOI: https://doi.org/10.7554/eLife.41444.015

**Figure supplement 2.** UPF1 association with nascent transcripts might depend on 5' splice sites recognition.
DOI: https://doi.org/10.7554/eLife.41444.016

abundant and fairly evenly distributed throughout the cytoplasm, as would be expected for mature mRNA, and there was relatively little in the nuclei (*Figure 6A*, panels I-III). By contrast, the nuclei of UPF1-depleted cells retained a substantial amount of poly(A), and the cells appeared to contain less cytoplasmic poly(A) than wild-type cells (*Figure 6A*, panels IV-VI). Much of the nuclear-retained poly (A) in the UPF1-depleted cells formed large cluster(s) in either inter-chromosomal spaces (*Figure 6A*, panel VI, white arrow) or, possibly more frequently, in surrounding nucleoli (yellow

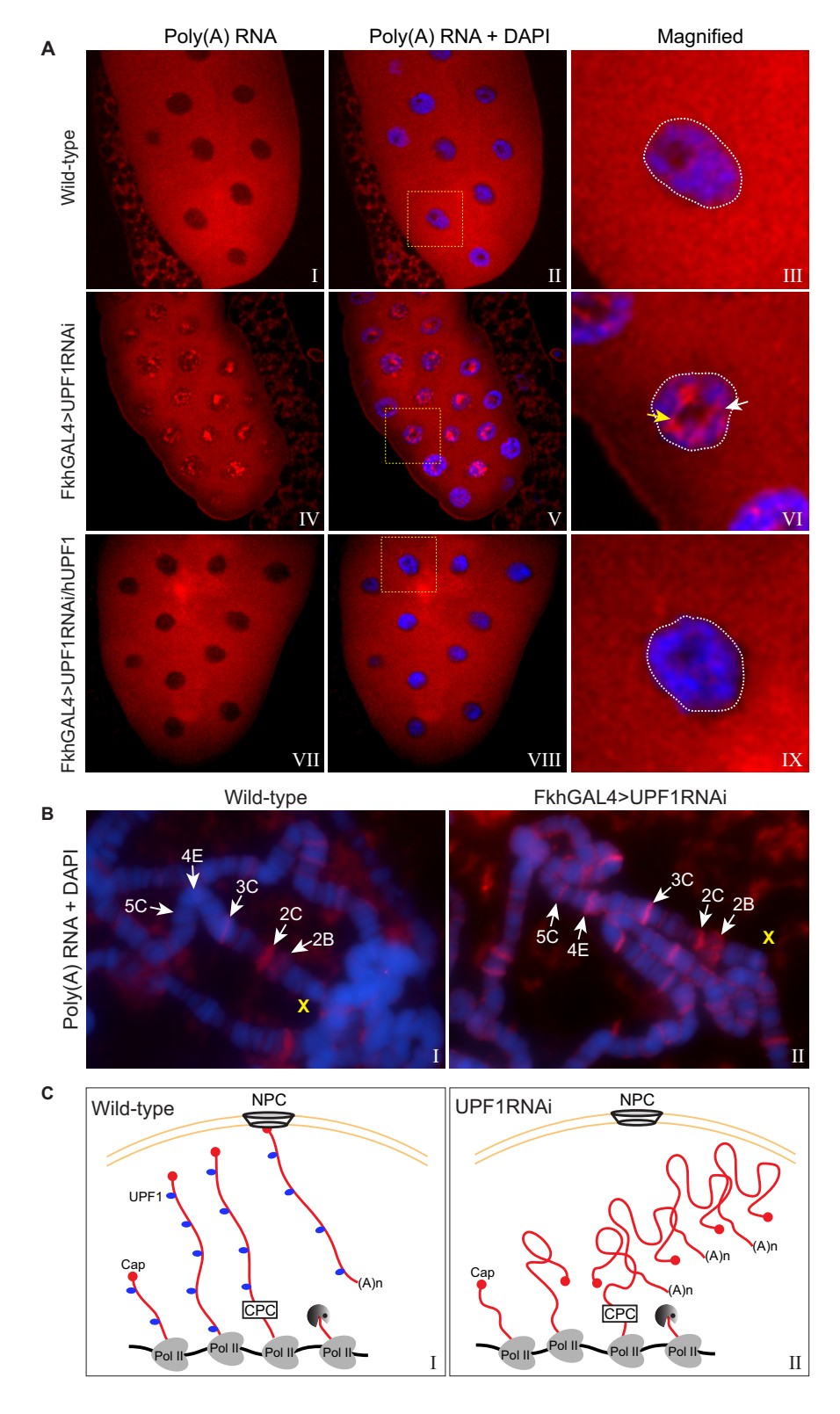

**Figure 6.** UPF1 knockdown results in nuclear accumulation and transcription sites retention of poly(A) mRNA. (**A**) Fluorescence in situ hybridisation (FISH) with a rhodamine-labelled oligo(dT)45 probe of third instar larval salivary glands, from either wild-type (top panel), UPF1-RNAi (middle panel) or UPF1-RNAi glands expressing human UPF1 (hUPF1, bottom panel). Chromosomes were counterstained with DAPI (blue). White arrow and yellow arrow indicate interchromosomal and perinucleolar aggregates respectively. (**B**) Oligo(dT)45 FISH (as above) of third instar larval salivary gland polytene

*Figure 6 continued on next page*

*Figure 6 continued*

chromosomes from either wild type or UPF1-RNAi. Chromosomes were counterstained with DAPI (blue). (C) Proposed model of accumulation of newly transcribed poly(A) mRNA at the site of transcription in UPF1 depleted cells (right) compared with wild type (left). Abbreviations: NPC for nuclear pore complex and CPC for cleavage and polyadenylation complex.

DOI: https://doi.org/10.7554/eLife.41444.017

arrow) that seemed neither linked to or in the proximity of any specific chromosomal region(s) or defined transcription site(s). In salivary glands over-expressing a transgene encoding human UPF1 there was no such nuclear poly(A) accumulation (*Figure 6A*, panels VII-IX); indicating that this phenotype is a direct consequence of the absence of UPF1. Additionally, no poly(A) accumulation was observed in UPF3[1] mutant glands, or following RNAi depletion of the EJC components Y14 or MAGO (unpublished data). Expression of hUPF1 also suppressed the apparent smaller size of the salivary glands depleted of the endogenous UPF1; these glands were comparable in size to that of wild-type (*Figure 6A*, panels I and VII vs IV).

An appreciable amount of poly(A) signal, which was not within clusters, was clearly at the chromosomes though, in the UPF1-depleted cells (*Figure 6A*, panels III and VI). We therefore used oligo (dT) FISH on polytene chromosome spreads to compare wild-type and UPF1-depleted cells and to assess whether there is retention of poly(A) near transcription sites. There was little poly(A) associated with most of the wild-type chromosomes. However, a few interbands – such as 2C at the distal end of the X chromosome (*Figure 6B*) – showed clear poly(A) signals (*Figure 6B*, left panel), suggesting that some mature mRNAs that have been cleaved and polyadenylated remain associated, at least briefly, with some transcription sites. Additionally, since UPF1 was obviously not associated with 2C (see *Figure 3A*), the poly(A) accumulation at 2C in wild-type cells may be a consequence of lack of UPF1 at this transcription site.

Both the number of transcriptional sites showing poly(A) accumulation and the amount of poly(A) RNA associated with these sites were strikingly increased in UPF1-depleted cells (*Figure 6B*, right panel). For example, there was no visible poly(A) accumulation at site 5C, which corresponds to the highly transcribed *Actin5C* gene, in wild-type, but this band was obviously fluorescent in UPF1-depleted cells. Another example was site 2B – where constitutively expressed *sta* and *rush* are probably the most active genes at this larval stage – which showed a faint poly(A) signal in wild-type glands and a strong signal in UPF1-depleted cells. UPF1 was clearly associated with these transcription sites (2B and 5C) on polytene chromosomes (*Figure 3A*) and in S2 cells (as detected by ChIP: see *Supplementary file 2* and *Figure 5—figure supplement 1* for the UPF1 profile of *Actin5C*).

Cumulatively, the data indicate that UPF1 plays important role(s) in the release of mRNAs from transcription sites and in their transport out of the nucleus (*Figure 6C* shows a cartoon of a transcription site in either a wild-type or UPF1 depleted cell with or without mRNA retention).

## Discussion

The RNA helicase UPF1 is typically most abundant in the cytoplasm and is mainly discussed in relation to NMD, leading to the common assumption that it acts mainly on mRNPs that have been exported from the nucleus. In contrast, we present evidence that UPF1 moves constantly within and between cell compartments, and that this shuttling depends on the ATPase activity of the RNA helicase domain, which is required for the dynamic binding and dissociation of UPF1 from mRNAs in both yeast and mammalian cells (*Bhattacharya et al., 2000*; *Franks et al., 2010*). Notably, UPF1 ATPase domain resides within a region previously thought to contain an atypical nuclear localisation signal in mammalian cells, which when deleted, abolished nuclear UPF1 accumulation following LMB treatment (*Mendell et al., 2002*). However, in view of UPF1 ATPase activity being required for this shuttling, we propose that typical UPF1 subcellular distribution, with more of the protein present in the cytoplasm than in the nucleus, is primarily a consequence of its continued association with abundant RNA cargos that are being continuously exported out of the nucleus. The NES-dependent CRM1 export pathway is not a major mRNA nuclear export mechanism in *Drosophila* nor in other well-studied organisms (*Herold et al., 2003*; *Hutten and Kehlenbach, 2007*), yet it is required for the export of vastly more abundant cargos consisting of RNP particles like ribosomal subunits and the signal recognition particle (SRP) (*Hutten and Kehlenbach, 2007*). Our data indicate that UPF1

might bind with SRP, as it strongly associated with the two loci encoding the 7SL RNA (*Figure 4—figure supplement 1B*), which make the SRP scaffold. However, UPF1 may not be binding the individual 40S and 60S subunits, as less UPF1 is detected in the centre of the nucleolus than in the remainder of the nucleus - discussed further below.

Within the nucleus we found UPF1 associated with many actively transcribing Pol II sites, to which it seems mainly to be recruited by an interaction with nascent pre-mRNA. More of the transcript-tethered UPF1 is associated with exons than with flanking introns. Splicing might enhance UPF1 binding, particularly with downstream exons (*Figure 5—figure supplement 2B*, Model 1), however, since UPF1 also associates with intron-less genes, splicing may not be necessary for its loading on nascent transcripts. The mechanism responsible for UPF1 being associated more with exons than with introns remains therefore to be elucidated. Possibly, UPF1 associates with all Pol II nascent transcripts, but on intron-containing pre-mRNAs, splice site recognition interferes with intron binding. Consistent with this interpretation, in the two genes we examined (*Xrp1* and *Socs36E*), the exon vs. intron bias in UPF1 binding is lost in cells depleted of the spliceosome component U1 snRNP. This suggests that, when U1 snRNP is bound to the 5'ss of an intron at the initial stage of splicing, it may hinder UPF1 translocation along the pre-mRNA and cause it to dissociate (see the Model two we propose in *Figure 5—figure supplement 2B*). UPF1 scanning of pre-mRNAs may therefore influence 5' splice sites recognition and affect splicing directly; changes in the relative concentrations of many alternatively spliced transcripts have been reported in UPF1-depleted S2 cells (*Brooks et al., 2015*).

Simple affinity of UPF1 for RNA is not likely to be the primary reason why UPF1 associates with some nascent transcripts, for several reasons: UPF1 does not associate with some highly transcribed Pol II genes, such as spliceosomal snRNAs; nor with snRNA U6 or other highly active Pol III genes; nor with rRNA genes transcribed by Pol I; there would be no differential affinity for introns *vs.* exons within a transcript; and UPF1 appears to be excluded from the RNA-packed centre of the nucleolus where rRNA genes are transcribed and ribosomal subunits are assembled (*McLeod et al., 2014*). What features of some nascent transcripts, most often of Pol II-transcribed genes, dictate that UPF1 becomes associated with them remain to be determined. One possible candidate would be the 7-methylguanosine (m7G) cap that is added co-transcriptionally to the 5'end of pre-mRNAs but not to Pol I and Pol III transcripts (*Ghosh and Lima, 2010*). Moreover, the m7G caps added to snRNAs and other small non-mRNA Pol II transcripts are further modified though by hypermethylation to generate 2,2,7-trimethylguanosine(m(3)G) structures (*Mouaikel et al., 2002*). Although this hypermethylation occurs after the transcripts have been exported to the cytoplasm in mammalian cells, perhaps this modification may occur in the nucleus instead in other organisms (*Mouaikel et al., 2002*), and this might explain why these classes of transcripts are not associated with UPF1 in *Drosophila*.

The association of UPF1 with nascent transcripts seems to be dynamic, and its putative 5'-to-3' scanning along RNA is likely to be fast and, at least on intron-containing pre-mRNAs, discontinuous. This pattern also suggests that when it encounters a steric block that cannot be removed, UPF1 must be capable of quickly dissociating and re-loading elsewhere on the transcript. In vitro, UPF1 can translocate along RNAs over long distances – but only at a maximum scanning velocity of ~80 base/min (*Fiorini et al., 2015*), which is much slower than the 2–3 kb/min of Pol II (*Fiorini et al., 2015*; *Fukaya et al., 2017*). Therefore, UPF1 either translocates along RNA faster in vivo, or its scanning is not processive as envisaged, or it is factually piggybacking on another entity that is capable of translocating on the RNA faster than UPF1. Although this machinery is unlikely to be the Pol II itself, as we found no evidence of a strong direct association of Pol II with UPF1.

Notably, we observed that less of UPF1 is associated with the 5' proximal region of nascent transcripts, which, in some instances, coincides with the 5'UTR (see *Actin5C* in *Figure 5—figure supplement 1B*). 5'UTRs are defined by the process of translation initiation, in which, following association with the mRNA's 5' cap, the 40S (carrying the initiator tRNA) migrates downstream until it recognises the start codon (*Kozak, 1989*). Less association of UPF1 with 5'UTRs might therefore signify that its binding to nascent transcripts is translation dependent, at least in some instances, and supports the view that ribosomes start scanning mRNAs cotranscriptionally; evidence of which has previously been reported in *Drosophila* (*Al-Jubran et al., 2013*; *Brogna et al., 2002*).

The most striking effects of UPF1 depletion were retention of poly(A) RNA at transcription sites and then its failure to be exported effectively from the nucleus. Mature mRNAs that have been cleaved and polyadenylated are normally expected to be speedily released from transcription sites, but our data show that this is not always the case. We have both: a) identified some sites on

polytene chromosomes that apparently accumulate poly(A) RNA even in wild-type glands; and b) shown that most of the active Pol II genes accumulate poly(A) RNA in UPF1-depleted salivary glands. Poly(A) RNA accumulation in UPF1-depleted cells is most prominent at genes with which UPF1 associates strongly in wild-type, such as *Actin5C* (shown both microscopically and by ChIP-seq). Conversely, those few transcription sites at which poly(A) accumulates even in wild-type cells, may not normally be associated with UPF1, a striking example of which is the 2C transcription site on the polytene chromosomes, where poly(A) accumulation was most apparent, but no obvious UPF1 association was observed.

Evidence of retention of poly(A) and specific mRNAs in discrete nuclear foci or 'dots' has previously been reported in cells defective in RNA processing, initially in mRNA export and processing mutants in yeast (*Jensen et al., 2001*); and later in other cells in which one of several RNA processing reactions are impaired (*Abruzzi et al., 2006*; *Paul and Montpetit, 2016*), including *Drosophila* cells carrying mutation in the RNA helicase P68 (*Buszczak and Spradling, 2006*). Whether the previously described 'dots' correspond to the poly(A) clusters that accumulate in the inter-chromosomal spaces of UPF1-depleted nuclei and/or to accumulations of poly(A) at transcription sites, which we identified here, remains to be determined.

In summary, our results indicate that UPF1 plays an important genome-wide role in the release of mRNAs from transcription sites and their export to the cytoplasm, at least in *Drosophila* salivary gland cells. Possibly, in the absence of UPF1 function, mRNPs acquire or retain native conformations that hinder their release from the chromosome and make them prone to aggregation and, consequently, cause nuclear retention. Such a global role of UPF1 could explain, better than its involvement in NMD, why this protein is universally conserved in eukaryotes, why its depletion affects the expression of a large fraction of the genome, and possibly why expression of human UPF1 in rat models of amyotrophic lateral sclerosis (ALS) overcomes the pathology caused by over-expression or mutation of the RNA-binding protein TDP-43 (*Barmada et al., 2015*; *Jackson et al., 2015*).

## Materials and methods

### Antibodies

The following antibodies were used for immunostaining: mouse anti-UPF1 (described as 7B12, 7D17 and 1C13 in this paper, typically diluted 1:100), mouse IgM anti-Ser2 Pol II (AB_10143905, H5, Covance, 1:500), mouse anti-hnRNPA1 (Hrb87F, P11, 1:50) (*Hovemann et al., 1991*), mouse anti-GFP (AB_627695, B-2, Santa Cruz, 1:200), Tetramethylrhodamine Conjugate Wheat Germ Agglutinin (Thermo Fisher, W7024, 10 μg/mL). The antibodies used in western blotting: mouse anti-UPF1 (7B12, 1:1000), mouse anti-α-tubulin (AB_477579, Sigma- Aldrich, 1:2500), rat anti-Ser2 Pol II (AB_11212363, Merck Millipore, 1:5000), mouse anti-hnRNPA1 (P11, 1:200), rabbit anti-eIF4AIII (1:1000), rabbit anti-Y14 (1:1000); the last two antibodies were described previously (*Choudhury et al., 2016*). The antibodies used in ChIP are mouse anti-UPF1 (7B12, see below, 5–10 μg), rabbit anti-Ser2 Pol II (AB_304749, Abcam, ab5095, 5 μg), mouse anti-Pol II (AB_306327, Abcam, ab817, 5 μg) and mouse anti-GFP (AB_627695, B-2, Santa Cruz, 5 μg).

### Drosophila stocks

Flies were reared in standard corn meal fly food media at 24°C. The y w[1118] strain was used as wild type (DGGR_108736). *UAS-UPF1-RNAi* (43144) and *UAS-GFP-UPF1* (24623) were obtained from the Bloomington stock centre. The *forkhead* (*Fkh*) Gal4 has a salivary gland specific expression from early stage of development (*Henderson and Andrew, 2000*). The UPF3[1] mutant was previously described (*Avery et al., 2011*). The transgenes expressing the *lacO*-tagged and ecdysone inducible S136 construct was described before (*Choudhury et al., 2016*). The transgene expressing human UPF1 (UAS-hUPF1) was generated by cloning the cDNA encoding wild-type human UPF1 into the EcoRI and NotI restriction sites in pUAST vector (*Brand and Perrimon, 1993*); after sub-cloning it from pCI-neo-hUPF1, previously described (*Sun et al., 1998*) into the NheI and SpeI sites in pBluescript. The *UAS-UPF1-RNAi* targets the following sequence located in the middle region of Drosophila UPF1: CCGGTTGTTATGTGCAAGAAA, which is significantly divergent in human UPF1 (CCAGTGGTGATGTGCAAGAAG) so that it cannot be targeted by this RNAi construct, as demonstrated in Results.

## Cell culture, RNA interference and Transfection

S2 cells (CVCL_Z232) were cultured in Insect–XPRESS media (Lonza) supplemented with 10% Fetal Bovine Serum (FBS) and 1% Penicillin-Streptomycin-Glutamine mix (P/S/G, Invitrogen) at 27°C. These cells never tested mycoplasma positive. To make the RNAi constructs for UPF1, eIF4AIII, Y14 and snRNPU1-70K mRNA, the specific sequences were PCR amplified from S2 cell genomic DNA, using corresponding primer pairs (*Supplementary file 3*). Along with the desired gene sequence, all the primer pairs carried the T7 promoter sequence (in bold) at their 5' end (5'-T**TAATACGACTCACTATAG**GGGAGA-3'). The amplified PCR fragments were purified using Monarch PCR and DNA Cleanup Kit (T1030S, NEB) and dsRNA was synthesised using the T7 RiboMAX express RNAi system (P1700, Promega). To induce RNAi, a six-well culture dish was seeded with $10^6$ cells/well in serum-free media and mixed with 15 µg of dsRNA/well. Following 1 hr incubation at RT, 2 mL of complete media was added to each well and the cells were incubated for the next three days to knockdown the corresponding RNA and then harvested. The RNAi efficiency of UPF1, eIF4AIII and Y14 was measured by western blotting while snRNPU1-70K was measured by real time PCR.

The four plasmids (B306, pAGW-N-term-GFP-UPF1; B307, pAc-C-term-GFP-UPF1; B309, pAGW-N-term-GFP-UPF1(DE-AA); and B310, pAc-C-term-GFP-UPF1(DE-AA) expressing *Drosophila* UPF1 tagged with GFP were generated by inserting the coding region of either wild-type UPF1 or UPF1 (DE617AA) in either pAc (GFP at the C-terminal) or pAGW (GFP at the N-terminal) Gateway compatible vector carrying a Act5C promoter sequence, as previously described (The Drosophila Gateway Vector collection, Carnegie Institution for Science). The coding region was PCR amplified from a full length Drosophila UPF1 cDNA clone previously described (*Brogna, 2000*); and the mutation was inserted using the QuikChange site-directed mutagenesis (Stratagene). Transfections were typically performed using TransIT-2020 transfection reagent (MIR5400) and cells were incubated for 24 hr at 27°C before further usage. For Leptomycin B (LMB) treatment, the cells were incubated with 50 nM LMB for 1 hr at RT.

## Generation of monoclonal antibodies against Drosophila UPF1

Antigens design, preparation, mice immunisation and hybridoma generation were carried out by Abmart (Shanghai). Twelve peptide sequences predicted to be highly immunogenic were selected from *D. melanogaster* UPF1 (*Supplementary file 1*) and cloned in-frame into an expression vector to produce a recombinant protein incorporating all 12 antigens which were used as the immunogen (Abmart, SEAL[TM] technology). Hybridoma clones were generated and used to induce 18 ascites, which were then screened by western blotting of S2 cell protein extracts. Out of these, three that showed a single band of the expected size and minimal cross-reactivity were selected and more of the monoclonal antibodies were subsequently purified from the corresponding hybridoma cell culture in vitro. Unless otherwise specified, 7B12 was used as the anti-UPF1 antibody throughout this study.

## Larval tissue immunostaining

Whole-mount immunostaining was performed as previously described (*Choudhury et al., 2016*). In brief, the internal organs of third instar larvae were dissected in 1X PBS (13 mM NaCl, 0.7 mM Na2HPO4, 0.3 mM NaH2PO4, pH 7.4) and fixed in 4% formaldehyde for 20 min at RT. Tissues were washed in 1XPBS followed by 1% Triton X-100 treatment for 20 min. Tissues were washed and incubated in blocking solution (10% Fetal Bovine Serum (FBS), 0.05% Sodium Azide in 1X PBS) for 2 hr at RT and then incubated in primary antibodies at 4°C overnight. Tissues were washed and further incubated with appropriate fluorescent-tagged secondary antibodies for 2 hr, typically. After washing, tissues were incubated in DAPI (4–6-diamidino-2-phenylindole, Sigma-Aldrich, 1 µg/mL) for 10 min and mounted in PromoFluor Antifade Reagent (PK-PF-AFR1, PromoKine) mounting medium and examined using a Leica TCS SP2-AOBS confocal microscope.

## LMB, DRB and larvae heat shock treatment

Wandering third instar larvae were dissected in M3 media and tissues were incubated with or without Leptomycin B (LMB, 50 nM) for 1 hr at RT. To examine the real-time effect of LMB treatment in the living cell, salivary glands were dissected in M3 media and incubated with a hanging drop of 50 nM LMB in M3 media in a cavity slide (*Singh and Lakhotia, 2015*). The fluorescence signal was

acquired at 5 min intervals with a Leica TCS SP2-AOBS confocal microscope. For ecdysone treatment, salivary glands were dissected in M3 media and incubated in 20-hydroxyecdysone (Sigma-Aldrich, H5142, 1 µM) for 1 hr at RT. For RNase treatment, salivary glands were dissected in M3 media and incubated in 0.1% Triton X-100 for 2 min prior to adding RNase A (Invitrogen, 100 µg/mL) and performing 1 hr incubation at RT. To examine the effect of 5, 6-Dichlorobenzimidazole 1-β-D-ribofuranoside (DRB) treatment, salivary glands were dissected in M3 media and incubated with DRB (Sigma-Aldrich, 125 µM) for 1 hr at RT. For heat shock response, larvae were placed in a pre-warmed microfuge tube lined with moist tissue paper and incubated in water-bath maintained at 37 ± 1°C for 1 hr.

## Live cell imaging (FRAP and FLIP)

Fluorescence recovery after photobleaching (FRAP) and fluorescence loss in photobleaching (FLIP) methods have been previously described (*Klonis et al., 2002*). Salivary glands expressing GFP-UPF1 were dissected from third instar larvae and mounted as a hanging drop in M3 media. For the FRAP the region of interest (ROI, a circle of fixed diameter) was rapidly photobleached with 100 iterations of 100% power Argon laser (488 nm) exposure. Subsequent recovery of fluorescence in the photobleached region was examined at defined time intervals. Fixed cells were examined as a control to confirm irreversible photobleaching. FRAP experiments were carried out on salivary glands at room temperature. The fluorescence signal in ROI was normalised and data analysed following published methods (*Phair and Misteli, 2000*; *Singh and Lakhotia, 2015*). FLIP experiments were done as previously described (*Phair and Misteli, 2000*). Following acquisition of five control images, GFP fluorescence in ROI1 was continuously photobleached with Argon laser (488 nm) at 100% power by 50 iterations. The loss in fluorescence in another region of interest, the ROI2 was measured for the same length of time. Fluorescence intensities at ROI1 and ROI2 were normalised and data analysed as described (*Nissim-Rafinia and Meshorer, 2011*). Both photobleaching experiments have been done using a Leica TCS SP2-AOBS confocal microscope.

## Polytene chromosomes immunostaining

Apart from the changes detailed below, the procedure was mostly as previously described (*Rugjee et al., 2013*). Briefly, actively wandering third instar larvae were dissected in 1X PBS and salivary glands were first fixed with 3.7% formaldehyde in 1X PBS and then with 3.7% formaldehyde in 45% acetic acid for 1 min each (*Singh and Lakhotia, 2012*). For Pol II immunostaining, salivary glands dissected in 1XPBS were incubated directly with 3.7% formaldehyde in 45% acetic acid for 3 min. Salivary glands were squashed in the same solution under the coverslip. Slides were briefly dipped in liquid nitrogen, the coverslips were flipped off with a sharp blade and then immediately immersed in 90% ethanol and stored at 4°C. For immunostaining, the chromosomes were air dried and then rehydrated by incubating the slide with 1XPBS in a plastic Coplin jar. Chromosomes were incubated in blocking solution (as for the tissue immunostaining) for 1 hr at RT and then incubated with primary antibodies diluted in blocking solution in a humid chamber overnight at 4°C. Chromosomes were washed in 1X PBS three times and further incubated with appropriate fluorescent-tagged secondary antibodies diluted in blocking solution for 2 hr at RT in the humid chamber. After washing, chromosomes were counterstained with DAPI and mounted in PromoFluor mounting media. Chromosomes were examined under Nikon Eclipse Ti epifluorescence microscope, equipped with ORCA-R2 camera (Hamamatsu Photonics).

## Fluorescent Oligo (dT) in situ hybridisation (FISH)

Oligo (dT) FISH was done as previously described for mammalian cells with some modifications (*Tripathi et al., 2015*). Salivary glands of third instar larvae were dissected in 1XPBS and fixed in 4% formaldehyde for 15 min at RT. Glands were then washed in 1XPBS and incubated with 0.1% Triton X-100 with 1 U/µL Ribolock RNase Inhibitor (ThermoFisher Scientific, EO0381) in 1XPBS for 10 min on ice and then rinsed further with 1XPBS three times with 5 min intervals and with 2XSSC for 10 min. Salivary glands were incubated with 5 ng/µL rhodamine-labelled oligo(dT)45 probe (IDT) in hybridisation solution (25% Formamide, 2X SSC pH 7.2, 10% w/v Dextran sulfate (Sigma Aldrich), 1 mg/mL *E. coli* tRNA (Sigma-Aldrich, R1753) for 12 hr at 42°C. Glands were then washed with freshly made wash buffer (50% Formamide in 2XSSC pH 7.2), followed by 2XSSC, 1XSSC and finally with

1XPBS three times each, with 5 min interval. Nuclei were counterstained with DAPI and tissues were mounted in PromoFluor Antifade mounting medium and examined under Leica TCS SP2-AOBS confocal microscope.

For polytene chromosomes oligo(dT) FISH, salivary glands were dissected in 1XPBS and incubated with fixing solution (1.85% formaldehyde in 45% acetic acid) for 5 min at RT. Chromosomes were squashed in the same solution and examined immediately under phase-contrast microscope to check if properly spread. Slides with good chromosomes were briefly dipped in liquid nitrogen and the coverslips were flipped off with a sharp blade. Slides were immediately dipped in 90% alcohol and stored at 4°C. Before hybridisation, slides were air dried and rehydrated in 1XPBS and then washed and hybridised, as described above for whole salivary glands. Chromosomes were counterstained with DAPI, mounted in PromoFluor Antifade mounting medium and examined under Nikon Eclipse Ti epifluorescence microscope.

## Immunoprecipitation

Immunoprecipitation was performed as previously described (*Hintermair et al., 2016*), with some modifications as detailed below. S2 cells ($4 \times 10^7$) were harvested and washed with ice-cold 1X PBS containing 1X PhosSTOP (Roche, 04906845001) and 1X cOmplete, Mini, EDTA-free Protease Inhibitor Cocktail (Roche, 04693159001). Cells were incubated in the hypotonic AT buffer (15 mM HEPES pH 7.6, 10 mM KCl, 5 mM MgOAc, 3 mM CaCl2, 300 mM Sucrose, 0.1% Triton X-100, 1 mM DTT, 1X PhosSTOP, 1X cOmplete, Mini, EDTA-free Protease Inhibitor Cocktail and 1 U/µL Ribolock RNase Inhibitor) for 20 min on ice and lysed with 2 mL Dounce homogenizer by 30 strokes with the tight pestle. Lysate was centrifuged at 5000 RPM for 5 min at 4°C in microcentrifuge and the nuclear pellet was resuspended in 500 µL IP buffer (50 mM Tris-HCl pH 8.0, 150 mM NaCl, 1% NP-40 (Roche), 1X PhosSTOP, 1X cOmplete, Mini, EDTA-free Protease Inhibitor Cocktail, 1 U/µL Ribolock RNase Inhibitor) for 20 min on ice. Nuclear lysates were sonicated using a Bioruptor sonicator (Diagenode) for 3 cycles of 30 s ON and 30 s OFF with maximum intensity. Following sonication, the lysates were centrifuged at 13000 RPM for 15 min at 4°C in a microfuge and the antibody (5 µg) was added to the clear supernatant, with or without addition of RNase A (100 µg/mL), and incubated overnight at 4°C on a rocker. Following incubation, 20 µL of prewashed paramagnetic Dynabeads (ThermoFisher Scientific, 10004D) were added and incubated further for 2 hr at 4°C on a rocker. Beads were washed 5 times with IP buffer using a magnetic rack and proteins were extracted by adding 40 µL SDS-PAGE sample buffer.

## ChIP-Seq

S2 cells ($2 \times 10^7$) were harvested and fixed with 1% formaldehyde (EM grade, Polyscience) for 10 min at RT. Following fixation, cross-linking reaction was stopped by adding 125 mM Glycine for 5 min at RT. Cells were centrifuged at 2000 RPM for 5 min at 4°C, the pellet was washed twice with ice-cold 1X PBS containing 1X cOmplete, Mini, EDTA-free Protease Inhibitor Cocktail. The cell pellet was resuspended in 1 mL of cell lysis buffer (5 mM PIPES pH 8.0, 85 mM KCl, 0.5% NP-40) supplemented with 1X cOmplete, Mini, EDTA-free Protease Inhibitor Cocktail and 1X PhosStop and incubated for 10 min at 4°C. Cells were centrifuged and the pellet was resuspended in 1 mL nuclear lysis buffer (50 mM Tris pH 8.0, 10 mM EDTA, 1.0% SDS) supplemented with 1X cOmplete, Mini, EDTA-free Protease Inhibitor Cocktail and 1X PhosStop and incubated for 10 min at 4°C. The cell suspension was further diluted with 500 µL IP dilution buffer (16.7 mM Tris pH 8.0, 1.2 mM EDTA, 167 mM NaCl, 1.1% Triton X-100, 0.01% SDS) and sonicated for 5 cycles at 30 s ON, 30 s OFF at maximum intensity using a Bioruptor sonicator (Diagenode); this produced an average fragment size of ~500 bp. Samples were centrifuged at 13000 RPM for 20 min in a microcentrifuge and the clear supernatant was transferred to a 15 mL tube. An aliquot of 100 µL supernatant was kept to extract input DNA. The supernatant was further diluted with 5 vol of IP dilution buffer. For each ChIP, typically we added 5 to 10 µg of antibody to this supernatant and incubated overnight at 4°C on a rocker. Prewashed 20 µL Dynabeads were added to the lysate-antibody mix and incubated further for 1 hr at 4°C on a rocker. Beads were washed 6 times with low salt buffer (0.1% SDS, 1% Triton X-100, 2 mM EDTA, 20 mM Tris pH 8.0, 150 mM NaCl), once with high salt buffer (0.1% SDS, 1% Triton X 100, 2 mM EDTA, 20 mM Tris pH 8.0, 500 mM NaCl) and once with 1X TE buffer (10 mM Tris pH 8.0, 1 mM EDTA). The beads were then incubated with 250 µL elution buffer (0.1M NaHC03, 1% SDS) at RT for

15 min and eluted chromatin was reverse cross-linked by adding 38 μL de-crosslinking buffer (2M NaCl, 0.1M EDTA, 0.4M Tris pH 7.5) and then incubated at 65°C overnight on a rotator. Proteins were digested by adding 2 μL Proteinase K (50 mg/mL) and incubated at 50°C for 2 hr on a rocker. DNA was isolated using Monarch PCR and DNA Cleanup Kits. Real-time PCR quantification of DNA samples was carried out using the SensiFAST SYBR Hi-ROX Kit (Bioline, BIO-92005) in 96-well plates using an ABI PRISM 7000 system (Applied Biosystems). For NGS sequencing, ChIP and input DNA were further fragmented to 200 bp fragment size using a Bioruptor Pico (Diagenode). All ChIP-DNA libraries were produced using the NEBNext Ultra II DNA Library Prep Kit (New England Biolab E7645L) and NEBnext Multiplex Oligos for Illumina Dual Index Primers (New England Biolabs E7600S), using provided protocols with 10 ng of fragmented ChIP DNA. Constructed libraries were assessed for quality using the Tapestation 2200 (Agilent G2964AA) with High Sensitivity D1000 DNA ScreenTape (Agilent 5067–5584). Libraries were tagged with unique barcodes and sequenced simultaneously on a HiSeq4000 sequencer.

## Nascent RNA isolation from S2 cells

Nascent RNA isolation was performed as previously described (Khodor et al., 2011). Briefly, S2 cells ($4 \times 10^7$) were harvested and washed twice with ice-cold 1X PBS via centrifugation at 2000 g for 5 min each. Cells were resuspended in 1 mL ice-cold buffer AT and incubated on ice for 10 min. Cells were lysed using a 2 mL Dounce homogenizer by 30 strokes with the tight pestle. The lysate was divided into two aliquots and each aliquot of 500 μL was layered over a 1 mL cushion of buffer B (15 mM HEPES-KOH at pH 7.6, 10 mM KCl, 5 mM MgOAc, 3 mM CaCl2, 1 M sucrose, 1 mM DTT, 1X cOmplete, Mini, EDTA-free Protease Inhibitor Cocktail), and centrifuged at 8000 RPM for 15 min at 4°C in a microcentrifuge. The supernatant was removed and the pellet was resuspended in 5 volumes of nuclear lysis buffer (10 mM HEPES-KOH pH 7.6, 100 mM KCl, 0.1 mM EDTA, 10% Glycerol, 0.15 mM Spermine, 0.5 mM Spermidine, 0.1 M NaF, 0.1 M Na3VO4, 0.1 mM ZnCl2, 1 mM DTT, 1X cOmplete, Mini, EDTA-free Protease Inhibitor Cocktail and 1 U/μL Ribolock RNase Inhibitor) and resuspended using a 2 mL Dounce homogenizer by three strokes with loose pestle and two strokes with tight pestle. Equal volume of 2X NUN buffer (50 mM HEPES-KOH pH 7.6, 600 mM NaCl, 2 M Urea, 2% NP-40, 1 mM DTT, 1X cOmplete, Mini, EDTA-free Protease Inhibitor Cocktail and 1 U/μL Ribolock RNase Inhibitor) was added to the nuclear suspension drop by drop while vortexing and the suspension was placed on ice for 20 min prior to spinning at 13,000 RPM for 30 min at 4°C. The supernatant was removed and TRI Reagent (Sigma, T9424) was added to the histone–DNA-Pol II-RNA pellet. The TRI Reagent–pellet suspension was incubated at 65°C with intermittent vortexing to dissolve the pellet, and RNA was extracted following the manufacturer's protocol. Poly(A) depletion was performed with Dynabeads Oligo(dT)25 (ThermoFisher Scientific). The purification of nascent RNA was assessed by RT-PCR of CG12030, CG5059 and CG10802 genes which have slow rates of co-transcriptional splicing (Khodor et al., 2011); cDNA synthesis was performed using qScript cDNA synthesis kit (Quanta Biosciences, 95047–025).

## RNA-seq

Extracted RNA samples were quantified using a Nanodrop-8000 Spectrophotometer (ThermoFisher ND-8000-GL) to assess quality and to determine concentrations. Aliquots of each sample were diluted to ~5 ng/μl, and tested with an Agilent Tapestation 2200 (Agilent G2964AA) using High Sensitivity RNA ScreenTapes kit (Agilent 5067–5579) to determine the RNA Integrity Number.

Total-RNA (1 μg) was first poly(A) selected using the NEBNext Poly(A) mRNA Magnetic Isolation Module (New England Biolabs E7490L) prior to library construction. Nascent RNA samples (100 ng) were processed without poly(A) selection. RNA libraries were prepared using a NEBNext Ultra Directional RNA Library Prep Kit (New England Biolab E7420L) and NEBnext Multiplex Oligos for Illumina Dual Index Primers (New England Biolabs E7600S), following standard protocols. RNA libraries were checked for quality using the Tapestation 2200 (Agilent G2964AA) with High Sensitivity D1000 DNA ScreenTape (Agilent 5067–5584). Multiplexed libraries were sequenced (50 bp single-end reads) on a HiSeq4000 sequencer.

## CHIP-seq and RNA-seq data analysis

ChIP-seq and RNA-seq data were initially viewed and analysed using the Lasergene Genomics Suite version 14 (DNASTAR). Pre-processing, assembly and mapping of the sequencing reads in the FASTQ files were performed by the SeqMan NGen software of this package automatically after selecting the NCBI *D. melanogaster* Dm6 genome release and accompanying annotations. Assembly and alignment output files for each genome contig were then analysed with the ArrayStar and Gen-Vision Pro software (from the same package) to view and compare read profiles on the genome. Profiles at selected regions were saved as high-resolution images.

To perform the metagene analyses, an index for Dm6 was downloaded from the HISAT2 website. HISAT2 v2.1.0 was then used to align the FASTQ files on it. The resulting SAM files were converted to BAM format, sorted, and indexed with Samtools 1.6. For the cytoplasmic RNA-seq data, the NCBI RefSeq gene annotations for Dm6 were downloaded as a refGene table from UCSC Table Browser (genome.ucsc.edu). The LiBiNorm tool was then used to produce read counts per gene in an HTSeq-count compatible format based on the refGene file (*Dyer et al., 2019*). Transcript lengths were also obtained from the refGene file and used together with total mapped sequencing reads to convert counts into RPKM values. For both ChIP-seq and nascent RNA-seq data, the BAM files were converted to Bedgraph files. This was carried out with the genomeCoverageBed command and options -bga and -ibam from the Bedtools v2.26.0 suite. Custom Perl scripts were then used to filter the Dm6 annotations either for genes separated by a minimum distance to avoid overlapping signals or RNA-seq expression levels. Subsequently, custom scripts were used to extract the signal from the Bedgraph files for each entry in the filtered gene list. A single base resolution was used for flanking regions, while the signal in gene bodies was binned into 16 bins to take account of different gene lengths. Each dataset was normalised by the total mapped sequencing reads in that dataset. Cross-referencing between different datasets was done based on the 'name' field, after filtering the annotations for multiple entries with the identical name.

A custom script was used to extract the sequencing read coverage from the Bedgraph files for each exon/intron/exon region in the dm6 annotation file xon_fly_gene (downloaded from UCSC Table Browser). To normalise for any bias in the sequencing, the UPF1 signal of each exon or intron was divided by the average coverage in the input sample. The fold change of UPF1 signal/input signal in introns was compared to that of their flanking exons using Wilcox.test (two sided and unpaired) in R (www.r-project.org). This analysis was done using either all introns annotated in Dm6 (151623) or those longer than 100 bp (76708). In either case flanking exons are significantly more enriched than introns. Here we have shown the result of analysis using just the longer introns as these were considered to be more informative because of the predicted lower resolution of ChIP at discriminating between closely adjacent sequences and because of the lower sequencing coverage of shorter introns compared to longer introns. All ChIP-seq and RNA-seq raw sequencing data and Bedgraph files were deposited in the GEO repository (Accession No GSE116808). All custom scripts used in this study are provided in Source Code File 1.

## Acknowledgements

We thank Bob Michell for critically reading the manuscript and for many valuable discussions. Thanks also to Michael Rosbash and Michael Marr (Brandeis University) for providing a detailed Nascent RNA-seq protocol, Harald Saumweber (Humboldt University, Berlin) for the P11 antibody, Lynne Maquat (University of Rochester) for sharing the human UPF1 cDNA plasmid; Isabel Palacios (University of Cambridge) for sharing the UPF3[1] mutant; and Bloomington Drosophila Stock Center for providing fly stocks, and Drosophila Genomic Resource Center for plasmids. We would like to acknowledge Michael Ashburner's (Cambridge) support, in whose laboratory SB generated the transgenic flies expressing human UPF1 back during his PhD. Thanks also to Alessandro Di Maio and the Birmingham Advanced Light Microscopy (BALM) facility; our School Drosophila research community, Yun Fan for advice on fly genetics, the fly food facility and Shrikant Jondhale for fly stocks maintenance; our NGS facility; and, Mike Tomlinson for providing Odyssey infrared imaging system for western blot detection. We thank also Pawel Grzechnik for reagents, and his and our group for help and continuous discussions. Thank you Louise for proofreading the manuscript. This project was funded by a Leverhulme Trust (RPG-2014–291) and BBSRC (BB/M022757/1) project grants, and at its

start, Wellcome Trust (9340/Z/09/Z) to SB. DH was supported by BBSRC grants BB/M017982/1 and BB/L006340/1.

## Additional information

### Funding

| Funder | Grant reference number | Author |
|---|---|---|
| Leverhulme Trust | RPG-2014-291 | Saverio Brogna |
| Wellcome | 9340/Z/09/Z | Saverio Brogna |
| Biotechnology and Biological Sciences Research Council | BB/M022757/1 | Saverio Brogna |
| Biotechnology and Biological Sciences Research Council | BB/M017982/1 | Daniel Hebenstreit |
| Biotechnology and Biological Sciences Research Council | BB/L006340/1 | Daniel Hebenstreit |

The author declare that funders had no role in study design, data collection and interpretation, or the decision to submit the work for publication.

### Author contributions

Anand K Singh, Conceptualization, Resources, Data curation, Software, Formal analysis, Validation, Investigation, Visualization, Methodology, Writing—original draft, Writing—review and editing; Subhendu Roy Choudhury, Conceptualization, Resources, Validation, Methodology; Sandip De, Conceptualization, Resources, Data curation; Jie Zhang, Data curation, Software, Formal analysis; Stephen Kissane, Resources, Methodology, Writing—review and editing; Vibha Dwivedi, Marija Petric, Conceptualization, Formal analysis, Writing—review and editing; Preethi Ramanathan, Conceptualization, Resources, Methodology; Luisa Orsini, Resources, Supervision, Writing—review and editing; Daniel Hebenstreit, Data curation, Formal analysis, Supervision, Funding acquisition, Writing—review and editing; Saverio Brogna, Conceptualization, Resources, Data curation, Software, Formal analysis, Supervision, Funding acquisition, Investigation, Visualization, Writing—original draft, Project administration, Writing—review and editing

### Author ORCIDs

Anand K Singh (iD) http://orcid.org/0000-0001-6500-6727
Daniel Hebenstreit (iD) http://orcid.org/0000-0003-0144-6728
Saverio Brogna (iD) https://orcid.org/0000-0001-7063-4381

### Decision letter and Author response

Decision letter https://doi.org/10.7554/eLife.41444.030
Author response https://doi.org/10.7554/eLife.41444.031

## Additional files

### Supplementary files

• Source code file 1. Custom scripts used to analyse ChIP-seq and RNA-seq data.
DOI: https://doi.org/10.7554/eLife.41444.018

• Supplementary file 1. Peptides used for UPF1 antibody generation.
DOI: https://doi.org/10.7554/eLife.41444.019

• Supplementary file 2. Most enriched transcription units by UPF1 ChIP-seq.
DOI: https://doi.org/10.7554/eLife.41444.020

• Supplementary file 3. List of PCR primers used.
DOI: https://doi.org/10.7554/eLife.41444.021

• Transparent reporting form

DOI: https://doi.org/10.7554/eLife.41444.022

### Data availability

ChIp-Seq and RNA -Seq data have been deposited in GEO under accession code GSE116808, GSE116806 and GSE116807.

The following datasets were generated:

| Author(s) | Year | Dataset title | Dataset URL | Database and Identifier |
|---|---|---|---|---|
| Singh AK, Brogna S | 2019 | The RNA helicase UPF1 associates with mRNAs cotranscriptionally | https://www.ncbi.nlm.nih.gov/geo/query/acc.cgi?acc=GSE116808 | NCBI Gene Expression Omnibus, GSE116808 |
| Singh AK, Brogna S | 2019 | The RNA helicase UPF1 associates with mRNAs cotranscriptionally | https://www.ncbi.nlm.nih.gov/geo/query/acc.cgi?acc=GSE116806 | NCBI Gene Expression Omnibus, GSE116806 |
| Singh AK, Brogna S | 2019 | The RNA helicase UPF1 associates with mRNAs cotranscriptionally | https://www.ncbi.nlm.nih.gov/geo/query/acc.cgi?acc=GSE116807 | NCBI Gene Expression Omnibus, GSE116807 |

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
