## [Decision Letter]

[Editors’ note: the authors were asked to provide a plan for revisions before the editors issued a final decision. What follows is the editors’ letter requesting such plan.]

Thank you for sending your article entitled "The RNA helicase UPF1 associates with mRNAs co-transcriptionally and is required for the release of mRNAs from gene loci" for peer review at *eLife*. Your article is being evaluated James Manley as the Senior Editor, a Reviewing Editor, and two reviewers.

Given the list of essential revisions, including new experiments, the editors and reviewers invite you to respond with an action plan and timetable for the completion of the additional work. We plan to share your responses with the reviewers and then issue a binding recommendation.

Summary:

Singh et al. describe the isolation of monoclonal antibodies that recognize *Drosophila* UPF1 and then utilize those reagents (primarily a single antibody) and UPF1-GFP to examine the subcellular distribution of Upf1 in a very large number of experiments. Experimental approaches include immunofluorescence (IF) analyses of salivary glands and polytene chromosomes, ChIP-seq studies in S2 cells, RNAi knockdowns, and nuclear vs. cytoplasmic distributions in response to LMB treatment. Based predominantly on correlative data, the authors conclude that Upf1 appears to be associated with "most of the active Pol II transcription sites and some Pol III-transcribed genes…," i.e., their conclusions represent a large leap from the conventional notion of UPF1 (and the NMD pathway) as a cytoplasmic mRNA surveillance system.

Essential revisions:

The reviewers agree that the manuscript should be "streamlined" to eliminate the less compelling data represented in the last two figures. In addition, the manuscript relies heavily on the monoclonal antibodies, and these need to be characterized better. The key concerns are:

1) Uncertainty about the reliability of the antibody used for immunolocalization of UPF1.

2) Absence of any information about the features of nascent transcripts that provide mechanistic insight into their putative interaction with UPF.

Experiments are recommended that would address mechanistic depth are those in which the "binding" ability of mutated UPF1 is assessed. More precisely, the authors should test mutations in UPF1 known to affect the protein's ability to bind RNA. Such mutational analyses would be critical to determine whether there is bona fide UPF1:RNA interaction, or whether some other interaction (e.g., association with an RNA-bound protein) has led to the observed result. If these can be addressed to the satisfaction of the reviewers, we would consider a revised manuscript. However, you may prefer to submit to another journal if you feel these issues, including validation of the antibodies, could not be addressed in a few months.

*Reviewer #1:*

Singh and colleagues present here a large amount of interesting data revolving around UPF1 functions in the nucleus at the site of transcription and during mRNA export. Using immunostainings on *Drosophila* salivary gland chromosome spreads and in S2 cells, combined with ChIP-seq data, the authors confirm previous evidence that UPF1 is a shuttling protein that exchanges rapidly between cytoplasm and nucleus, with the majority of UPF1 residing in the cytoplasm. They further show that inside the nucleus, UPF1 localizes at sites of active transcription (primarily Pol II transcription sites), suggesting that UPF1 binds RNA co-transcriptionally. While the data for these findings is compelling and in my opinion would justify publication on its own (Figure 1, Figure 2, Figure 3, Figure 1—figure supplement 1, Figure 1—figure supplement 3, Figure 2, Figure 3— figure supplement 2, Figure 4—figure supplement 1 and Figure 4—figure supplement 2, Figure S3 and Figure S5), the authors went on with additional experiments and further analysis of their ChIP-seq data that led them to conclude (i) that intron recognition interferes with UPF1 association, (ii) that Pol II tends to stall near transcription start sites in UPF1-depleted cells and (iii) that UPF1 depletion leads to nuclear mRNA retention. While the effect of UPF1 depletion on poly(A)+ RNA is striking, the conclusions of points (i) and (ii) are much less compelling and would need further investigation to gain more mechanistic insight, as detailed below. I therefore suggest removing these two parts from the manuscript. If confirmed by additional data, these findings can be published later in separate papers.

*Reviewer #2:*

Singh et al. describe the isolation of monoclonal antibodies that recognize *Drosophila* UPF1 and then utilize those reagents (primarily a single antibody) and UPF1-GFP to examine the subcellular distribution of UPF1 in a very large number of experiments. Experimental approaches include immunofluorescence (IF) analyses of salivary glands and polytene chromosomes, ChIP-seq studies in S2 cells, RNAi knockdowns, and nuclear vs. cytoplasmic distributions in response to LMB treatment. Based predominantly on correlative data, the authors conclude that UPF1 appears to be associated with "most of the active Pol II transcription sites and some Pol III-transcribed genes…," i.e., their conclusions represent a large leap from the conventional notion of UPF1 (and the NMD pathway) as a cytoplasmic mRNA surveillance system. Although the authors' conclusions raise the possibility of unforeseen functions for UPF1 they are contingent on two key questions: (1) how reliable are the antibody-based experiments? and (2) what feature of nascent RNA transcripts might be responsible for their apparent interactions with UPF1? Unfortunately, neither question is addressed adequately.

Specificity of the anti-UPF1 monoclonal antibodies was considered by comparisons of IF results with those obtained with UPF1-GFP, and examination of the consequences of RNAi knockdown of UPF1. UPF1-GFP results were markedly different from those obtained in IF analyses (e.g., Figure 1, Figure 1—figure supplement 3, Figure S3, Figure S5), with the UPF1-GFP results largely showing cytoplasmic and (interesting) perinuclear staining without any significant nuclear staining, whereas the IF experiments seem to indicate that UPF1 can be found in all three locations. RNAi knockdown of UPF1 was only examined in salivary glands, where it depresses but does not eliminate anti-UPF1 staining of polytene chromosomes. Importantly, the authors did not present an experiment examining IF of S2 cells or mosaic tissues in which the UPF1 gene has been deleted. These critical experiments are a fairly commonplace approach to demonstrating unambiguous specificity of an antibody reagent. Other trivial controls that are lacking are results from an experiment using only the secondary antibody and IF localization of a protein definitively known to be cytoplasmic (using the same secondary antibody used for the UPF1 experiments).

Little insight is presented with regard to possible salient features of nascent transcripts that might mediate their putative interactions with UPF1. In light of known interactions between UPF1 and ribosomes it is surprising that the authors did not consider their own published work demonstrating that *Drosophila* ribosomal proteins harboring fluorescent tags manifest cytological distributions that are substantially similar to the putative UPF1 distributions seen in the current manuscript (Rugjee et al., 2013; Brogna et al., 2002). Clearly, one possibility is that the distributions seen in the present work are an indirect consequence of UPF1 interactions with chromosome-associated ribosomes or ribosomal proteins. Further, in light of UPF1's known role as an RNA-binding protein I would have expected that the authors would test whether UPF1 mutations known to impair the protein's RNA-binding capacity altered any of the cytological or ChIP-seq results seen by the authors.

[Editors' note: formal revisions were requested, following approval of the authors’ plan of action. A further plan for revisions was requested prior to acceptance, as described below.]

Thank you for sending your article entitled "The RNA helicase UPF1 associates with mRNAs co-transcriptionally and is required for the release of mRNAs from gene loci" for peer review at *eLife*. Your article is being evaluated by James Manley as the Senior Editor, a Reviewing Editor, and two reviewers.

Given the list of essential revisions, including new experiments, the editors and reviewers invite you to respond with an action plan and timetable for the completion of the additional work. We plan to share your responses with the reviewers and then issue a binding recommendation.

Both reviewers have concerns regarding the work, although to different degrees. They have discussed this extensively between them. Reviewer 1 is concerned about the interpretation of the ChIP data against the suggestion that it be removed or at least qualified that UPF1 could bind after intron removal. In any case, this observation does not seem to shed much light on the process.

Reviewer 2 judges that, since the interpretation of the data hinges on the specificity of the antibodies, there needs to be more rigorous demonstration that the antibodies have been well characterized and indirect effects have been ruled out. The reviewer states, "The current version of the manuscript continues to rely on RNAi knockdowns to substantiate MAb reliability, an approach that is less convincing than mosaics." and further "Most importantly, the paper is comprised of a series of observations without definitive experiments that rule out indirect interactions as explanations for the observed results."

In order to satisfy the reviewers, it appears that you will need to deliver a more definitive manuscript that addresses these concerns. Please let us know how you would plan to respond to the mentioned concerns.

*Reviewer #1:*

The authors have provided additional data to demonstrate the specificity of their antibodies, as requested.

As suggested be the reviewers, they also have streamlined the manuscript by removing the less compelling parts on a suggested role of UPF1 in Pol II pausing and at the nuclear envelope. Against the reviewer's suggestion, they decided to retain their ChIP-seq experiments showing that more UPF1 is associated with exons than with introns. They argue that my alternative – and less exciting – interpretation of their data, namely that this might simply result from a substantial fraction of UPF1 binding only after intron removal, was unlikely to be valid. However, I do not understand why the fact that UPF1 was also found associated with intronless genes would challenge my interpretation of the data. With ChIP, one has no information at which stage during the transcription of an intronless gene UPF1 binds the nascent transcript. The same is true for intron-containing genes and if a fraction of UPF1 would bind after an intron was spliced out, the net result would be more UPF1 bound to exons than to introns.

The newly added rescue of the poly(A)+ RNA accumulation in the nucleus in the absence of UPF1 demonstrates the specificity of this observation and strengthens the conclusion.

Thus, even though I do not agree with the author's interpretation of the ChIP data, all my major concerns have been satisfactorily addressed and I support accepting the manuscript for publication.

*Reviewer #2:*

Singh et al. have submitted a revised version of their manuscript that bears a close resemblance to the original, both in experimental approach and in the concerns raised by the respective experiments and conclusions. As before, the authors begin by describing the isolation of monoclonal antibodies that recognize *Drosophila* UPF1 and then utilize those antibodies (primarily monoclonal antibody 7B12) and UPF1/GFP constructs to examine the subcellular distribution of UPF1 in S2 cells and salivary glands. Additional experiments evaluated ChIP-seq in S2 cells, and nuclear vs. cytoplasmic distributions in response to LMB treatment. Based predominantly on correlative data, the authors conclude that UPF1 appears to be associated with most of the active Pol II transcription sites and some Pol III-transcribed genes. As such, their conclusions represent a large leap from the conventional notion of UPF1 (and the NMD pathway) as a cytoplasmic mRNA surveillance system. Although the authors' conclusions raise the possibility of unforeseen functions for UPF1, they are contingent on two key questions: (1) how reliable are the IF, ChIP-seq, and GFP-UPF1 experiments? and (2) is it possible that the observed UPF1:RNA interactions are indirect?

Previously, I was concerned about the 7B12 antibody, particularly: (i) the very strong additional band that showed up in the westerns of Figure 1—figure supplement 1B and (ii) the lack of any tests of the antibody using UPF1 deletion/mosaic tissues. The current version of the manuscript continues to rely on RNAi knockdowns to substantiate MAb reliability, an approach that is less convincing than mosaics. Further, I find the vast overexpression of UPF1-GFP (~10x the level of endogenous UPF1), and its distribution differences relative to endogenous UPF1 to be additional concerns. In the ChIP-seq experiments, far too much attention is paid to hypothetical explanations for the differences in exon vs. intron signals (a difference that could simply be accounted for by co-transcriptional splicing) while little attention was paid to the ~10x lower recovery of UPF1 ChIP-seq vs. Pol II ChIP-seq.

Most importantly, the paper is comprised of a series of observations without definitive experiments that rule out indirect interactions as explanations for the observed results. The authors only make that concern worse by filling the discussion with numerous unsubstantiated explanations for the apparent shuttling, the ChIP-seq results, and the experiments with the UPF1 helicase mutant.

In short, this manuscript just isn't up to *eLife*'s standards.

---

## [Author Response]

[Editors' note: the authors’ plan for revisions was approved and the authors made a formal revised submission.]

Essential revisions:The reviewers agree that the manuscript should be "streamlined" to eliminate the less compelling data represented in the last two figures. In addition, the manuscript relies heavily on the monoclonal antibodies, and these need to be characterized better. The key concerns are:1) Uncertainty about the reliability of the antibody used for immunolocalization of UPF1.

As explained in detail in our responses to reviewer 1 below, we are very confident that the three UPF1 antibodies detect specific signals in all the assays we have performed. Prompted by this concern though, we have redone all the controls and most of the additional experiments suggested. The results of these confirm that the antibodies are specific and we show the additional data in new supplementary figures/panels in the revised manuscript: (Figure 1—figure supplement 2A and Figure 3—figure supplement 1C).

2) Absence of any information about the features of nascent transcripts that provide mechanistic insight into their putative interaction with UPF.Experiments are recommended that would address mechanistic depth are those in which the "binding" ability of mutated UPF1 is assessed. More precisely, the authors should test mutations in UPF1 known to affect the protein's ability to bind RNA. Such mutational analyses would be critical to determine whether there is bona fide UPF1:RNA interaction, or whether some other interaction (e.g., association with an RNA-bound protein) has led to the observed result. If these can be addressed to the satisfaction of the reviewers, we would consider a revised manuscript. However, you may prefer to submit to another journal if you feel these issues, including validation of the antibodies, could not be addressed in a few months.

We have done more experiments as suggested, to assess whether UPF1 RNA “binding” ability affects its nuclear localization. We examined this in S2 cells transfected with GFP tagged constructs expressing either wild-type UPF1 or a mutant version which carries a well characterized mutation in the ATPase domain that inhibits its RNA helicase activity, which is required for UPF1 dynamic binding and dissociation from mRNAs. The results showed the ATPase mutant version is not retained in the nucleus following incubation with LMB, unlike wild-type which readily accumulates in the nucleus (shown in the new Figure 2). In the revised Discussion section we therefore proposed that this rapid shuttling of UPF1 between nucleus and cytoplasm primarily depends on UPF1 binding to RNP cargos that are being continuously exported out of the nucleus, rather than on a direct or protein adapter-mediated interaction with CRM1, as previously suggested (referred in the manuscript). A more detailed molecular analysis of how UPF1 interacts with nascent mRNPs will need to be done in future, but we feel that this would go beyond what we aimed to demonstrate with this initial study – see our more detailed response to reviewer 1’s criticism about this point below.

We agree that it is plausible that a substantial fraction of UPF1 might associate indirectly with nascent transcripts through interactions with RNPs loaded on them. Following the reviewer suggestion, we have speculated what these RNPs and mechanisms might be in the revised Discussion section (see our detailed response to the reviewer below).

As advised, we have “streamlined” the manuscript. All the data that suggested a role of UPF1 in Pol II pausing were omitted (old Figure 5 and S10); these data remain public on the bioRxiv preprint and we hope to investigate this further and publish them at a later stage separately, as advised by reviewer 1. We have also omitted the data suggesting a role of UPF1 at the nuclear envelope (old Figure S3B).

We have kept the data showing that more UPF1 is associated with exons than with introns (old Figure 4A-D). Although we agree that more data will be required to gain insight in to the mechanism, this exon vs. intron bias in the ChIP-seq data is genome-wide and apparent at several genes that we have verified by qPCR (see our detailed response to reviewer 1 about this point). However, we simplified the figure by moving three of the panels to a supplementary figure (new Figure 5—figure supplement 2) and hope this change will make the manuscript both more concise and clear.

With regards to our conclusion that UPF1 plays a role in the release of mRNAs from transcription sites and export to the cytoplasm, as suggested by reviewer 1, we have tested whether the striking poly(A) accumulation phenotype observed in UPF1 depleted cells can be rescued by expression of a UPF1 transgene, which is not targeted by UPF1-RNAi. The result was a clear-cut rescue of this phenotype, which, hence, provides further indication that nuclear poly(A) accumulation is a direct effect of UPF1 depletion (discussed further in response to this reviewer).

Reviewer #1:Singh and colleagues present here a large amount of interesting data revolving around UPF1 functions in the nucleus at the site of transcription and during mRNA export. Using immunostainings on Drosophila salivary gland chromosome spreads and in S2 cells, combined with ChIP-seq data, the authors confirm previous evidence that UPF1 is a shuttling protein that exchanges rapidly between cytoplasm and nucleus, with the majority of UPF1 residing in the cytoplasm. They further show that inside the nucleus, UPF1 localizes at sites of active transcription (primarily Pol II transcription sites), suggesting that UPF1 binds RNA co-transcriptionally. While the data for these findings is compelling and in my opinion would justify publication on its own (Figure 1, Figure 2, Figure 3, Figure 1—figure supplement 1, Figure 1—figure supplement 3, Figure 2, Figure 3— figure supplement 2, Figure 4—figure supplement 1 and Figure 4—figure supplement 2, Figure S3 and Figure S5), the authors went on with additional experiments and further analysis of their ChIP-seq data that led them to conclude (i) that intron recognition interferes with UPF1 association, (ii) that Pol II tends to stall near transcription start sites in UPF1-depleted cells and (iii) that UPF1 depletion leads to nuclear mRNA retention. While the effect of UPF1 depletion on poly(A)+ RNA is striking, the conclusions of points (i) and (ii) are much less compelling and would need further investigation to gain more mechanistic insight, as detailed below. I therefore suggest removing these two parts from the manuscript. If confirmed by additional data, these findings can be published later in separate papers.

As suggested, we have removed all these data from the revised manuscript; these remain public in the initial bioRxiv preprint (395863).

As for the data indicating a link between intron recognition and UPF1 association with nascent transcripts (old Figure 4 and Figure 5—figure supplement 1), we think that this is perhaps one of the most exciting parts of our observations. This was indeed a completely unexpected result, and, as the reviewer commented, difficult to believe, initially also by us. However, our UPF1 ChIP-seq data clearly indicate that more UPF1 is associated with exons than with introns genome wide, and this observation was validated with multiple ChIP experiments and by real-time PCR of different genes (for example *Xrp1*, new Figure 5B). Irrespective of what the mechanism responsible might be, we think this striking exon-biased enrichment needs to be discussed in the manuscript. As introns and exons are defined co-transcriptionally, such an enrichment profile is consistent with the rest of the data and our conclusion that UPF1 association with nascent transcripts/RNPs is what primarily drives recruitment to transcription sites. Although the road-block model that we proposed (old Figure 4G, now new Figure 5—figure supplement 2B) is largely a working model, as we made it clear in the manuscript, it is the best explanation we can give presently for why less UPF1 is associated with the intron than the upstream flanking exon. The reviewer argues instead that a more likely interpretation for our results is that a substantial fraction of UPF1 binds the RNA after intron removal. We considered this possibility, but several of our observations argue against it. First, UPF1 is also associated with intronless genes, for example at the heat-shock genes such as 87A and 87C on the polytene chromosomes (new Figure 3B), and Gapdh1 in S2 cell as detected by ChIP (Figure 5—figure supplement 1). Second, UPF1 was also associated with our intronless transgene upon ecdysone induction (new Figure 3C shows the intron-less version). Third, a ChIP enrichment profile such as that of *Xrp1* (new Figure 5A and 5B) strongly suggests that UPF1 should be able to get released from exons as transcription progresses. Contrary to the reviewer’s prediction, the overall UPF1 ChIP-seq signal does not visibly change between the last three exons of *Xrp1* (new Figure 5A), and more significantly, UPF1 association does not, on average, increase toward the 3’ end of genes (see metagene in new Figure 4D).

Reviewer #2:Singh et al. describe the isolation of monoclonal antibodies that recognize Drosophila UPF1 and then utilize those reagents (primarily a single antibody) and UPF1-GFP to examine the subcellular distribution of UPF1 in a very large number of experiments. Experimental approaches include immunofluorescence (IF) analyses of salivary glands and polytene chromosomes, ChIP-seq studies in S2 cells, RNAi knockdowns, and nuclear vs. cytoplasmic distributions in response to LMB treatment. Based predominantly on correlative data, the authors conclude that UPF1 appears to be associated with "most of the active Pol II transcription sites and some Pol III-transcribed genes…," i.e., their conclusions represent a large leap from the conventional notion of UPF1 (and the NMD pathway) as a cytoplasmic mRNA surveillance system. Although the authors' conclusions raise the possibility of unforeseen functions for UPF1 they are contingent on two key questions: (1) how reliable are the antibody-based experiments? and (2) what feature of nascent RNA transcripts might be responsible for their apparent interactions with UPF1? Unfortunately, neither question is addressed adequately.Specificity of the anti-UPF1 monoclonal antibodies was considered by comparisons of IF results with those obtained with UPF1-GFP, and examination of the consequences of RNAi knockdown of UPF1. UPF1-GFP results were markedly different from those obtained in IF analyses (e.g., Figure 1, Figure 1—figure supplement 3, Figure S3, Figure 3—figure supplement 1), with the UPF1-GFP results largely showing cytoplasmic and (interesting) perinuclear staining without any significant nuclear staining, whereas the IF experiments seem to indicate that UPF1 can be found in all three locations. RNAi knockdown of UPF1 was only examined in salivary glands, where it depresses but does not eliminate anti-UPF1 staining of polytene chromosomes. Importantly, the authors did not present an experiment examining IF of S2 cells or mosaic tissues in which the UPF1 gene has been deleted. These critical experiments are a fairly commonplace approach to demonstrating unambiguous specificity of an antibody reagent. Other trivial controls that are lacking are results from an experiment using only the secondary antibody and IF localization of a protein definitively known to be cytoplasmic (using the same secondary antibody used for the UPF1 experiments).

Although the relative nuclear level of GFP-UPF1 (in the earlier version we wrongly referred to as UPF1-GFP) is not as high as endogenous UPF1 in salivary glands (this is probably a consequence of the fact that GFP-UPF1 is highly overexpressed) its overall pattern is similar to that of the endogenous protein: the signal is mostly cytoplasmic but also apparent in the region of the nucleus occupied by the chromosomes, and absent from the centre of the nucleolus. Moreover, as demonstrated by the LMB treatment, UPF1-GFP is constantly shuttling between nucleus and cytoplasm. As for the authenticity of the perinuclear signal, though this is most obvious with GFP-UPF1, it was also detected by immunostaining of the endogenous UPF1. This is apparent in some of the other cell types shown (Figure 1—figure supplement 3). In the revised manuscript, to address the concern about the antibody specificity, we have shown additional immunostainings of normal and UPF1-depleted salivary glands that demonstrates the veracity of the perinuclear signal of endogenous UPF1, with any of the three antibodies and, consistent with it being specific, depleted by UPF1-RNAi (this is particularly apparent in the images shown in the new Figure 1—figure supplement 2). Please also see our response to reviewer 1 about the specificity of the antibody.

Little insight is presented with regard to possible salient features of nascent transcripts that might mediate their putative interactions with UPF1. In light of known interactions between UPF1 and ribosomes it is surprising that the authors did not consider their own published work demonstrating that Drosophila ribosomal proteins harboring fluorescent tags manifest cytological distributions that are substantially similar to the putative UPF1 distributions seen in the current manuscript (Rugjee et al., 2013; Brogna et al., 2002). Clearly, one possibility is that the distributions seen in the present work are an indirect consequence of UPF1 interactions with chromosome-associated ribosomes or ribosomal proteins. Further, in light of UPF1's known role as an RNA-binding protein I would have expected that the authors would test whether UPF1 mutations known to impair the protein's RNA-binding capacity altered any of the cytological or ChIP-seq results seen by the authors.

We agree that a substantial fraction of UPF1 might be recruited indirectly on the nascent transcripts, by, for example, an interaction with associated RNPs. We agree with the reviewer that, as controversial as this explanation may be, these could be ribosomal subunits loaded on the nascent transcript (the presence of which we reported in the two papers mentioned and another which was not mentioned (Al-Jubran et al., 2013). Prompted by the reviewer’s comment, we discussed this possibility in the revised manuscript and explained our reasoning for thinking that simple affinity of UPF1 for RNA is not likely to be the primary reason why UPF1 associates with nascent transcripts. For example, UPF1 does not associate with rRNA genes, spliceosomal snRNAs, and most highly active Pol III genes. What is/are the putative mechanisms linking ribosome/ribosomal subunits association and UPF1 association with nascent transcripts will need to be addressed in future (translation initiation on nascent transcripts is our working hypothesis, as suspected by the reviewer, and have mentioned it in revised manuscript).

As for the suggestion to test UPF1 mutants, we performed additional experiments in S2 cells transfected with different variants of UPF1, including mutations (DE-AA) in its RNA helicase domain. The results of these experiments strongly indicate that UPF1 RNA helicase activity is in fact required for its nucleocytoplasmic shuttling, and that it is possibly the primary determinant of its localisation in the cell where UPF1 distribution mirrors that of the mRNA, which is characteristically more abundant in the cytoplasm than in the nucleus. This new data have been added as a new figure (new Figure 2), described in the revised manuscript and their implications discussed in detail. Notably, as we have referred to in the new Discussion section, the DE-AA mutation that inactivates the ATPase domain resides within the region previously thought to function as a non-canonical “nuclear localization signal” in mammalian cells, as UPF1 lacks a clear NLS, as mentioned by the reviewer below.

[Editors’ note: what follows is the authors’ plan to address the revisions. This plan for revisions was approved and the authors made a formal revised submission.]

Both reviewers have concerns regarding the work, although to different degrees. They have discussed this extensively between them. Reviewer 1 is concerned about the interpretation of the ChIP data against the suggestion that it be removed or at least qualified that UPF1 could bind after intron removal. In any case, this observation does not seem to shed much light on the process.Reviewer 2 judges that, since the interpretation of the data hinges on the specificity of the antibodies, there needs to be more rigorous demonstration that the antibodies have been well characterized and indirect effects have been ruled out. The reviewer states, "The current version of the manuscript continues to rely on RNAi knockdowns to substantiate MAb reliability, an approach that is less convincing than mosaics." and further "Most importantly, the paper is comprised of a series of observations without definitive experiments that rule out indirect interactions as explanations for the observed results."In order to satisfy the reviewers, it appears that you will need to deliver a more definitive manuscript that addresses these concerns. Please let us know how you would plan to respond to the mentioned concerns.Reviewer #1:The authors have provided additional data to demonstrate the specificity of their antibodies, as requested.As suggested be the reviewers, they also have streamlined the manuscript by removing the less compelling parts on a suggested role of UPF1 in Pol II pausing and at the nuclear envelope. Against the reviewer's suggestion, they decided to retain their ChIP-seq experiments showing that more UPF1 is associated with exons than with introns. They argue that my alternative – and less exciting – interpretation of their data, namely that this might simply result from a substantial fraction of UPF1 binding only after intron removal, was unlikely to be valid. However, I do not understand why the fact that UPF1 was also found associated with intronless genes would challenge my interpretation of the data. With ChIP, one has no information at which stage during the transcription of an intronless gene UPF1 binds the nascent transcript. The same is true for intron-containing genes and if a fraction of UPF1 would bind after an intron was spliced out, the net result would be more UPF1 bound to exons than to introns.The newly added rescue of the poly(A)+ RNA accumulation in the nucleus in the absence of UPF1 demonstrates the specificity of this observation and strengthens the conclusion.Thus, even though I do not agree with the author's interpretation of the ChIP data, all my major concerns have been satisfactorily addressed and I support accepting the manuscript for publication.Reviewer #2:Singh et al. have submitted a revised version of their manuscript that bears a close resemblance to the original, both in experimental approach and in the concerns raised by the respective experiments and conclusions. As before, the authors begin by describing the isolation of monoclonal antibodies that recognize Drosophila UPF1 and then utilize those antibodies (primarily monoclonal antibody 7B12) and UPF1/GFP constructs to examine the subcellular distribution of UPF1 in S2 cells and salivary glands. Additional experiments evaluated ChIP-seq in S2 cells, and nuclear vs. cytoplasmic distributions in response to LMB treatment. Based predominantly on correlative data, the authors conclude that UPF1 appears to be associated with most of the active Pol II transcription sites and some Pol III-transcribed genes. As such, their conclusions represent a large leap from the conventional notion of UPF1 (and the NMD pathway) as a cytoplasmic mRNA surveillance system. Although the authors' conclusions raise the possibility of unforeseen functions for UPF1, they are contingent on two key questions: (1) how reliable are the IF, ChIP-seq, and GFP-UPF1 experiments? and (2) is it possible that the observed UPF1:RNA interactions are indirect?Previously, I was concerned about the 7B12 antibody, particularly: (i) the very strong additional band that showed up in the westerns of Figure 1—figure supplement 1B and (ii) the lack of any tests of the antibody using UPF1 deletion/mosaic tissues. The current version of the manuscript continues to rely on RNAi knockdowns to substantiate MAb reliability, an approach that is less convincing than mosaics. Further, I find the vast overexpression of UPF1-GFP (~10x the level of endogenous UPF1), and its distribution differences relative to endogenous UPF1 to be additional concerns. In the ChIP-seq experiments, far too much attention is paid to hypothetical explanations for the differences in exon vs. intron signals (a difference that could simply be accounted for by co-transcriptional splicing) while little attention was paid to the ~10x lower recovery of UPF1 ChIP-seq vs. Pol II ChIP-seq.Most importantly, the paper is comprised of a series of observations without definitive experiments that rule out indirect interactions as explanations for the observed results. The authors only make that concern worse by filling the discussion with numerous unsubstantiated explanations for the apparent shuttling, the ChIP-seq results, and the experiments with the UPF1 helicase mutant.In short, this manuscript just isn't up to eLife's standards.

Thank you very much for handling the reviewing of our manuscript. We are very pleased that reviewer 1 is now satisfied and supports its publication. Although they favor an alternative explanation for why more of UPF1 is associated with exons, we still think that the most plausible explanation is that it mostly associates with an exon prior removal of the downstream intron. However, we agree that a portion of UPF1 might bind to exons only after intron removal. We will discuss this possibility in the next revision and expand/modify our model of how UPF1 associates with nascent transcripts accordingly.

Reviewer 2’s remaining concern about the specificity of the antibodies puzzles us. Following the first review round, as requested we have performed many control experiments to address this concern, and we think that the results leave no doubt about the specificity of the antibodies. Reviewer 1 appears satisfied with this interpretation. Briefly, in the revision we have shown that all three monoclonal UPF1 antibodies produce similar immunostainings of both the polytene chromosomes (new Figure 3—figure supplement 1) and the salivary glands (Figure 1—figure supplement 2A), and UPF1-RNAi depletes all three antibodies signals in the cells which express the RNAi construct, but not in the adjacent fat body cells attached to the glands (new Figure 1—figure supplement 2A, indicated by the yellow arrows). Moreover, GFP-tagged UPF1 shows a similar subcellular distribution, and in particular a highly similar banding pattern on the polytene chromosomes (new Figure 3—figure supplement 1D).

In the latest assessment reviewer 2 points out that the 7B12 antibody detects an additional strong band in one of the Western blots (Figure 1—figure supplement 1B). Although this was not pointed out in their initial review, it was by reviewer 1 and we had addressed it in our previous response. In brief, this extra band was seen only in the initial screening where the blots were incubated with ascites produced with the 7B12 hybriodoma, this cross-reactivity is minimal when using purified IgG from hybridoma supernatants (see Figure 1A). As explained in the manuscript, we have used only purified antibody not ascites in all subsequent assays.

This reviewer states that will be more convincing to test the specificity of the antibodies using genetic mosaics tissues. We don’t think that such experiments will be more informative or cleaner than what we have already performed. UPF1 is required for growth and survival of cell clones in imaginal discs (the typical tissue where clone analysis are performed), as a consequence homozygous Upf1(-) clones can’t be easily recovered and consist of just a few cells, as previously reported (Metzstein and Krasnow, 2006).

As for whether a fraction of UPF1 might be recruited indirectly on the nascent transcripts, by, for example, an interaction with associated RNPs, we agree, and, as suggested by reviewer 2’s original criticism, we have discussed this possibility extensively in the revised manuscript. Strangely, now they criticise us for having done so. Moreover, irrespective of its loading mechanism, we think that the data fully supports the manuscript main conclusion that UPF1 associates with nascent transcripts.

In summary, given that reviewer 1 appears to be satisfied with antibodies’ specificity and the rest of the manuscript, we would like you to consider whether doing the substantial more work requested by reviewer 2 is required. We are very excited about these findings and think the association of UPF1 with nascent transcripts is evolutionally conserved.

References:

Metzstein, M.M., and Krasnow, M.A. (2006). Functions of the nonsense-mediated mRNA decay pathway in *Drosophila* development. PLoS genetics 2, e180.